# Mutations in respiratory complex I promote antibiotic persistence through alterations in intracellular acidity and protein synthesis

Bram Van den Bergh [1,2,3], Hannah Schramke[4], Joran Elie Michiels[1,2], Tom E. P. Kimkes[4], Jakub Leszek Radzikowski[4], Johannes Schimpf[5], Silke R. Vedelaar[4], Sabrina Burschel[5], Liselot Dewachter[1,2], Nikola Lončar[6], Alexander Schmidt [7], Tim Meijer[4], Maarten Fauvart[1,2,8], Thorsten Friedrich [5], Jan Michiels [1,2,9]✉ & Matthias Heinemann [4,9]✉

Antibiotic persistence describes the presence of phenotypic variants within an isogenic bacterial population that are transiently tolerant to antibiotic treatment. Perturbations of metabolic homeostasis can promote antibiotic persistence, but the precise mechanisms are not well understood. Here, we use laboratory evolution, population-wide sequencing and biochemical characterizations to identify mutations in respiratory complex I and discover how they promote persistence in *Escherichia coli*. We show that persistence-inducing perturbations of metabolic homeostasis are associated with cytoplasmic acidification. Such cytoplasmic acidification is further strengthened by compromised proton pumping in the complex I mutants. While RpoS regulon activation induces persistence in the wild type, the aggravated cytoplasmic acidification in the complex I mutants leads to increased persistence via global shutdown of protein synthesis. Thus, we propose that cytoplasmic acidification, amplified by a compromised complex I, can act as a signaling hub for perturbed metabolic homeostasis in antibiotic persisters.

[1] Centre of Microbial and Plant Genetics, Department of Molecular and Microbial Systems, KU Leuven Leuven, Belgium. [2] Center for Microbiology, Flanders Institute for Biotechnology, VIB, Leuven, Belgium. [3] Department of Entomology, Cornell University, Ithaca, NY, USA. [4] Molecular Systems Biology, Groningen Biomolecular Sciences and Biotechnology Institute, University of Groningen, Nijenborgh 4, Groningen, The Netherlands. [5] Molecular Bioenergetics, Institute of Biochemistry, Albert-Ludwigs-University of Freiburg, Freiburg im Breisgau, Germany. [6] Molecular Enzymology, Groningen Biomolecular Sciences and Biotechnology Institute, University of Groningen, Nijenborgh 4, Groningen, The Netherlands. [7] Proteomics Core Facility, Biozentrum, University of Basel, Basel, Switzerland. [8] imec, Leuven, Belgium. [9] These authors jointly supervised this work: Jan Michiels, Matthias Heinemann. ✉email: jan.michiels@kuleuven.be; m.heinemann@rug.nl

**B**acterial persistence describes the presence of phenotypic variants within an isogenic population that are transiently tolerant to otherwise lethal antibiotic therapy[1–4]. Such antibiotic-tolerant cells can endanger human health as they prolong therapy and can ultimately lead to relapsing infections[1,3,5,6]. In addition, tolerance levels are shown to increase during evolution under periodic antibiotic treatment[7–10] and such increased tolerance levels were shown to proceed and accelerate the development of genetic resistance[2,11–13].

Important advances towards understanding bacterial persistence have been made over the last years. First, toxin-antitoxin systems were found to be involved in persistence induction and awakening[14–21]. Second, the stringent response, a stress response that coordinates adaptation to stresses through altered levels of messenger nucleotides ppGpp and pppGpp (i.e. (p)ppGpp), has been linked to persistence via either increased levels of (p)ppGpp[22–24] or a plethora of pathways including the induction of the RpoS regulon[25,26], the inhibition of major processes such as DNA replication, transcription, and translation[27,28] and the activation of toxin-antitoxin systems[18,29,30].

A recently growing body of evidence suggests that also the 'metabolic state' is a highly important and general signal in the context of persistence[31,32]. First, a variety of metabolic genes affect the frequency of persister cells[33–37]. Second, metabolic perturbations like diauxic[28] and rapid nutrient shifts[26,38] and entry in stationary phase[3,39,40] induce persister cell formation, where certain rapid carbon shifts can force almost an entire population into an antibiotic-tolerant state in a metabolic flux-dependent manner[38]. Other indications for metabolic perturbations driving persistence include the observed decrease in ATP concentration[15,21,41–43] as well as other imbalances of the metabolic network such as very fast growth[44], enzyme saturation[45], or a perturbed TCA cycle and/or respiration[40,46,47]. Given the multitude of various metabolic perturbations leading to persistence, we hypothesize that one generic signal exists that triggers persistence upon perturbed metabolic homeostasis. Yet, it is unknown what this trigger is and how it initiates persister cell formation.

Here, we describe a mechanism for how different perturbations of metabolic homeostasis lead to persister cell formation. Through laboratory evolution of persistence and whole-genome, population-wide sequencing, we identified respiratory complex I as a key mutational target for increased persister formation upon entry into stationary phase, in model and pathogenic *E. coli* strains. Complex I mutants with increased persistence display compromised proton translocation leading to significant cytoplasmic acidification upon perturbations of metabolic homeostasis. We propose that acidification induces the persister state in two ways: through activation of the RpoS regulon and, upon more severe cytoplasmic acidification, through shutdown of protein synthesis. Our findings show how cytoplasmic acidification, amplified by a compromised complex I, acts as a central signaling hub to connect a perturbed metabolic homeostasis with persister cell formation.

## Results

### Evolution towards increased persistence leads to mutations in *nuo*.
To discover genetic determinants of persister cell formation upon perturbations of metabolic homeostasis, we exploited laboratory evolution selecting for increased persistence in an environment that cycles between dilution and regrowth to stationary phase and antibiotic treatments (Fig. 1a)[8,10,48]. In this work, we investigated the genomic changes that occurred in 37 populations of an *E. coli* lab strain that were previously evolved under such cycling environment with daily exposure to amikacin

(ten populations), kanamycin, tobramycin, or gentamicin (nine populations each)[8]. For comparison and generalization, we also included three populations of the uropathogen UTI89 that we evolved under daily exposure to amikacin in the current work, where we found 1000 to 10,000-fold increases in persistence, similar to the previously evolved populations of the lab strain (Supplementary Fig. 1a).

To detect mutations in these 40 *E. coli* populations, we applied deep, whole-genome sequencing on population samples at the endpoint of evolution after 5–10 days. Initially focusing on mutations present at, or above, a frequency of 5% (see "Methods"), we identified on average 3.15 mutations per population amounting to a total of 128 mutations across all populations (Supplementary Data 1; Supplementary Fig. 1b, c). Statistical analyses indicate that the evolved populations consist entirely of mutants (Supplementary Fig. 1d, e). Confirming the low mutational complexity as inferred from population-wide sequencing, genome sequences of randomly picked clones at endpoints only contained one or two mutations (Supplementary Table 1). Mutations are significantly enriched in coding regions of 29 genes, specifically targeting genes coding for inner-membrane proteins (Supplementary Fig. 1f–h). Several targets are repeatedly hit by identical mutations in different populations, and/or emerge as multiple alleles (Supplementary Fig. 1b, j–l).

Mutations are strongly enriched in the *nuo* operon, which is hit 41 times by 19 unique alleles (Fig. 1b, Supplementary Fig. 1b, i–k), with eight alleles reoccurring identically in as many as ten populations (Supplementary Data 1; Supplementary Fig. 1l). When including mutations below our initial frequency cutoff of 5%, 237 additional mutations were uncovered in the 29 previously identified target genes (Supplementary Data 2), showing a similar enrichment in the *nuo* operon (52 hits) with 28 newly identified minor alleles (Fig. 1b). As overall 38.3% of each evolved population consists of *nuo* alleles (Supplementary Fig. 1m), we find that the *nuo* operon is the major target of evolution towards high persistence in our laboratory evolution experiment selecting for increased antibiotic survival upon perturbation of metabolic homeostasis through entry into stationary phase.

### Mutations in *nuo* predominantly hit regions encoding transmembrane domains involved in proton translocation and are causal to high persistence.
The *E. coli nuo* operon consists of 13 genes coding for NADH/ubiquinone oxidoreductase (respiratory complex I). The complex serves as an entry point for electrons from NADH into the electron transport chain (ETC) (Fig. 2a)[49]. 87 out of the 93 identified mutations and 41 unique alleles are present in the genes *nuoAHJKLMN* of complex I. Notably, the structurally and functionally distinct part of the complex, encoded by *nuoBCDEFGI*, contains significantly fewer mutations (only six) (Supplementary Fig. 2a, b). NuoBCDEFGI forms the cytoplasmic arm that catalyzes NADH oxidation and that transports the electrons via a series of iron-sulfur clusters to the ubiquinone reduction site, 100 Å apart[50]. NuoAHJKLMN builds the membrane arm that translocates protons across the membrane, thus contributing to the electrochemical membrane potential, i.e. proton motive force (PMF)[51,52] (Fig. 2a). The mutations we found are primarily in the proton translocating units of the membrane arm and specifically hit codons of amino acids enriched in hydrophobic residues (83 times) and of those residues located in transmembrane helices (64 times) (Fig. 2c, Supplementary Fig. 2a, b). While many of the altered amino acids are in close proximity to regions involved in proton translocation (Fig. 2a inset, Supplementary Fig. 2c, d)[51–53], the mutations do not directly affect amino acids reported to be crucial for proton translocation, such as amino acids involved in long-range

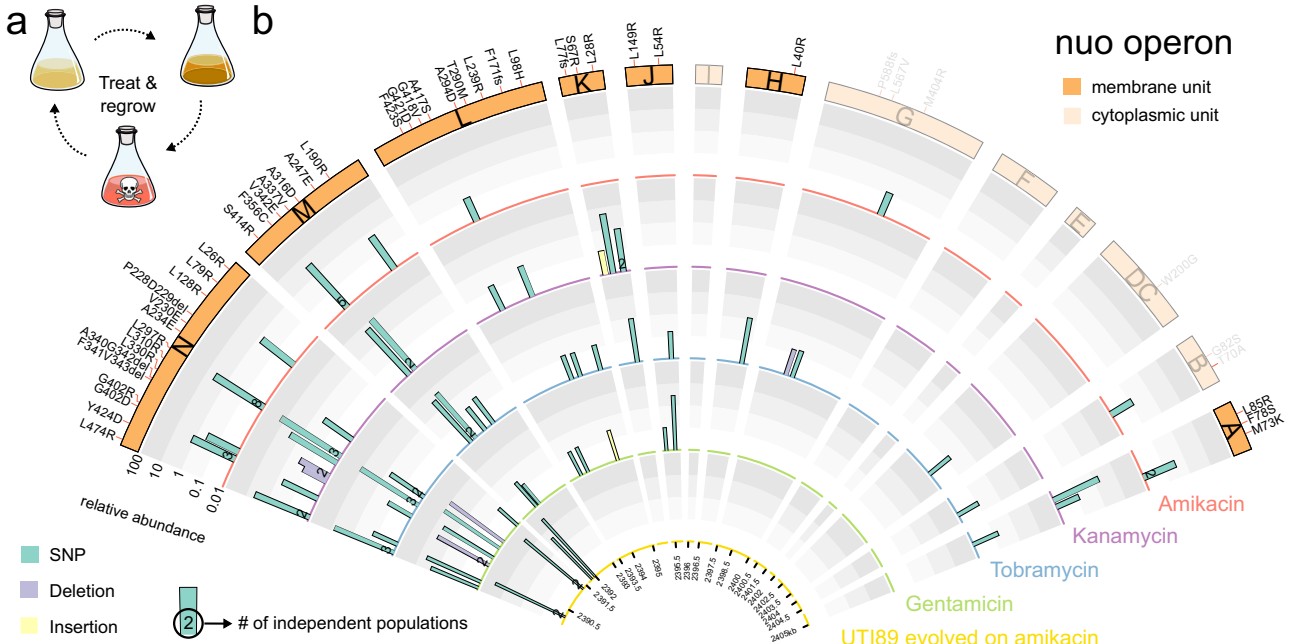

**Fig. 1 As main target of evolution in *E. coli* towards increased persistence, the *nuo* operon is hit by mutations in genes primarily encoding membrane-spanning units. a** Scheme of experimental evolution alternating daily between dilution and regrowth to stationary phase to a density of $1–5 \times 10^9$ CFU ml$^{-1}$ (respectively, light and dark orange) and antibiotic treatment (red), which selects for an increase in persistence. **b** A circos plot segment visualizes mutations that emerged in the evolved populations in genes of the *nuo* operon (orange rectangles with letters for each gene) which proved to be the main target of evolution (Supplementary Fig. 1). Genes encoding proteins that are part of the membrane-spanning arm of the complex and cytoplasmic domain are shown in bright and pale colors, respectively. Data of ten (amikacin), nine (kanamycin, tobramycin, and gentamicin) and three (UTI89) evolved populations were cumulated. Numbers in the bars show how many independently evolved populations contain that exact mutation (≥2). The concentric arcs show the relative frequency of mutations in populations evolved on different antibiotics (4 outer arcs; all treated for 5 h with 400 µg ml$^{-1}$) and in populations of the uropathogenic strain UTI89 evolved on amikacin (inner arc; for 5 h with 400 µg ml$^{-1}$). Target regions are rescaled for visual representation and different types of mutation are shown in different colors. The outer ring shows the resulting amino acid change at the protein level for each of the mutations. See also Supplementary Fig. 1, Supplementary Tables 1–2 and Supplementary Data 1–2. Source data are provided as a Source Data file.

structural changes along a central axis in the membrane arm, or in outlining the proton half-channels, or in the actual binding and export of protons (Fig. 2a). Based on these findings, we expected that the mutations do not abolish the overall complex I function, but rather induce smaller functional changes in complex I, likely connected with proton translocation, leading to the observed increase in persistence.

To confirm causality of the mutations in *nuo* for persistence, we selected clones with single mutations in each one of the subunits L, M and N (called *nuo\** further on)—which are subunits that are all proposed to contain one distinct proton translocation path[54]—and restored the chromosomal wild-type alleles in these mutants by site-specific mutagenesis. Here we found that the strains again became susceptible towards amikacin, i.e. the antibiotic that imposed selective pressure during evolution (Fig. 2d). In addition, the minimal inhibitory concentration of amikacin remained unchanged in the *nuo\** mutants, excluding an increase in resistance as an explanation for their increased survival (Supplementary Fig. 2e). In fact, the *nuo\** mutants show clear cross-tolerance to other antibiotics in the stationary phase (Supplementary Fig. 2f–h), which is a known hallmark of persistence[3]. As already expected from the above-described characteristics of the mutations (e.g., that they are specific to the proton translocation arm and are not targeting crucial amino acids), the effect of the point mutations cannot simply be mimicked by destroying complex I functionality entirely, either by knocking out the individual subunits or the entire complex (Fig. 2e, Supplementary Fig. 2f, g). Restoring the wild-type alleles abrogates not only the amikacin tolerance as mentioned above, but also the multidrug tolerance of the point mutants for the

related aminoglycoside kanamycin and for the unrelated fluoroquinolone ofloxacin (Supplementary Fig. 2i), demonstrating that *nuo\** mutations are causal to the high-persistence, multidrug-tolerant phenotype.

**High-persistence-conferring variants of complex I oxidize NADH and transfer electrons but are impaired in proton translocation.** To test whether the identified *nuo\** mutations indeed alter proton translocation, we produced and purified wild-type complex I and three variants, carrying a variation in either subunit L, M, or N. Despite its huge size (±550 kDa) and its multitude of cofactors, we were able to purify wild-type complex I and its variants with retained activity from *E. coli* membranes through several chromatographic steps (Supplementary Fig. 3a, b; see "Methods"). Yields of the preparations were similar between the samples (±0.2 mg protein g$^{-1}$ cells) as were the banding patterns after SDS PAGE, both in line with previously obtained results[55,56] (Supplementary Fig. 3c). Native gel electrophoresis confirmed stability and purity of the preparations, and in situ NADH oxidation demonstrated activity of the complexes (Supplementary Fig. 3d). The mutations did not introduce instability to the variants as judged from their "melting points"[57] (Supplementary Fig. 3e, f). Thus, the purification yielded pure and active samples of the wild-type complex I and the three variants.

To profile the biochemical activities of the purified complexes, we made use 1) of proteoliposomes, i.e. vesicles made from *E. coli* polar lipids reconstituted with the individual preparations of complexes, of membrane fractions containing the wild-type complex I or the variants, and 2) of inside-out vesicles (ISOVs)

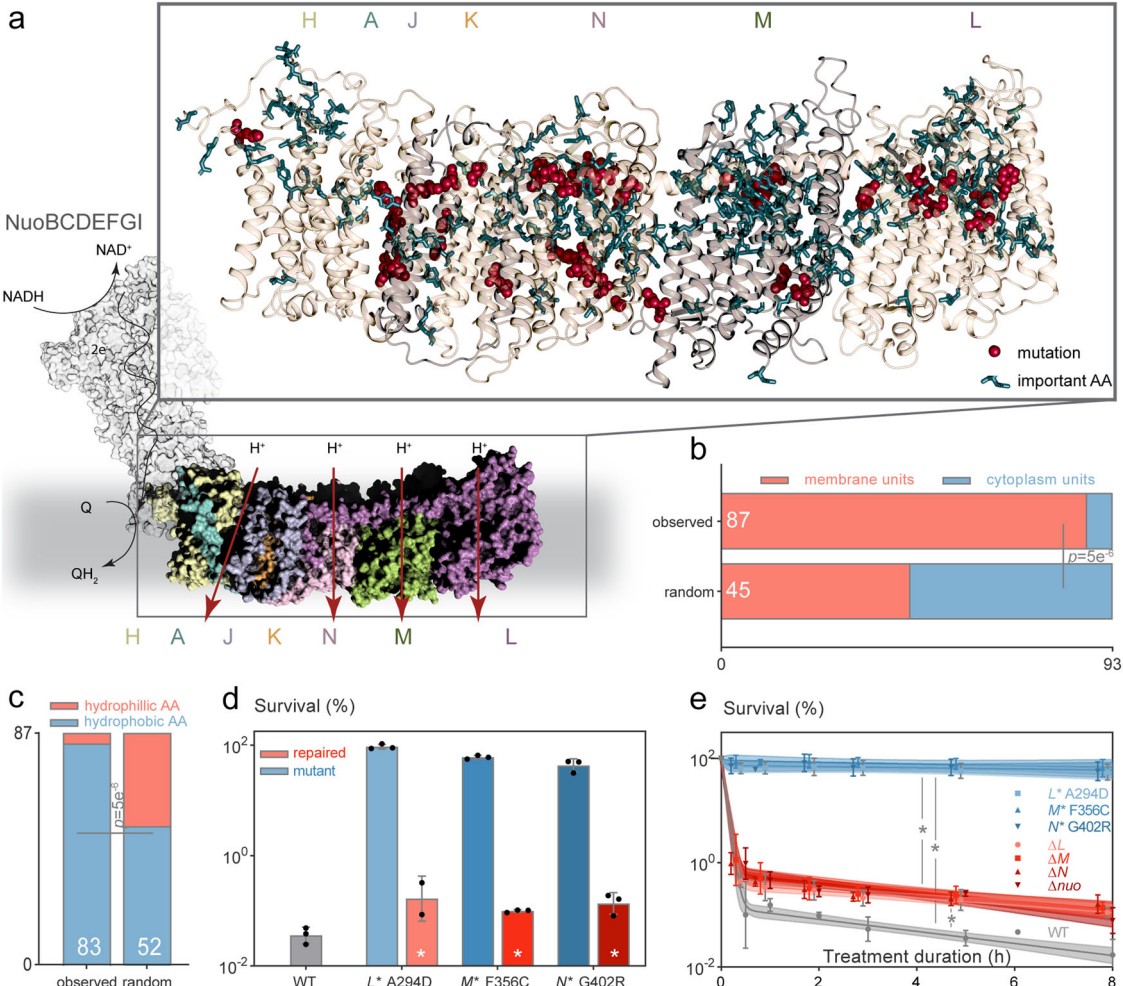

**Fig. 2 High persistence-conferring *nuo*\* mutations target hydrophobic amino acids in transmembrane subunits and do not cause a complete loss of function of the complex. a** Modeled structure of *E. coli* complex I (see Methods). The cytoplasmic subunits NuoBCDEFGI (gray surfaces) catalyze NADH oxidation and transfer two electrons over a series of iron-sulfur clusters (not shown) to the quinone reduction site. The quinol flows further down the electron transport chain in the membrane (gray background) to be re-oxidized by terminal oxidases (not shown). The hydrophobic subunits NuoAHJKLMN (in colored surfaces) translocate four $H^+$ ions per molecule of NADH that is oxidized by the cytoplasmic subunits. The membrane part is magnified in the inset and is annotated with the positions of amino acid variants found in high-persistence mutants (red spheres) and with residues that are crucial for proton translocation (blue sticks, based on refs. [51–53]). **b, c** High-persistence mutations are significantly enriched in **b** the membrane units and **c** predominantly target hydrophobic amino acids (Chi² comparison to random mutations, see Methods). **d** The *nuo*\* mutations are causal for high persistence. Mutants with single mutations in each one of subunits L, M and N lose their high tolerance for amikacin (5h with 400 µg ml⁻¹) when *nuo*\* mutations are repaired (mean ±stdevs, n = 3; \**p* < 0.0001 for a within-strain comparison from a two-way ANOVA with Šidák's posttest). **e** Killing dynamics with amikacin (400 µg ml⁻¹) confirm the high persistence of *nuo*\* point mutants in stationary phase and show that their effect cannot be mimicked by a single gene or operon knockout (in red). A model describing biphasic killing dynamics (lines ±95% CIs) was fitted to the data (means ±stdevs, n = 3; \* fits are different based on AIC criterion). See also Supplementary Fig. 2. Source data are provided as a Source Data file.

prepared from *E. coli* membrane fractions (see "Methods"). Like proteoliposomes, the latter are closed compartments but additionally retain the full complexity of the in vivo membrane composition. NADH oxidation (and subsequent reduction of decyl ubiquinon in the membrane) was measured spectrophotometrically as a decrease in NADH absorption at 340 nm, where we found that the liposomes reconstituted with wild-type complex I and the variants oxidized NADH at a similar rate (Fig. 3a and inset). Electrons from NADH ultimately flow down the ETC via the quinone/quinol couple to the terminal oxidases, where in well-aerated conditions oxygen is reduced to water. Respirometry on membranes that contain wild-type complex I or its variants as sole NADH dehydrogenase (see "Methods") revealed highly similar $O_2$ reduction rates upon NADH addition (Fig. 3b and inset). Thus, NADH oxidation and electron transfer are both not affected by the *nuo*\* mutations.

To examine proton translation as third functionality of complex I, we made use of the fluorescence of 9-amino-6-chloro-2-methoxyacridine (ACMA) that can be quenched upon alkalization induced by proton translocation of complex I, allowing to investigate differences in proton pumping (Supplementary Fig. 3g, h). Compared to wild-type complex I, all variants showed, to varying degrees (50–75%), impaired proton translocation upon addition of NADH to the proteoliposomes (Fig. 3c). We confirmed this result by testing the activity of the variants in ISOVs (Fig. 3d). The ISOVs results furthermore quantitatively agreed with their more complex composition. As ISOVs also contain the terminal quinol oxidases, proton gradients measured with ACMA reflect the combined action of the quinol oxidases (1H⁺/e⁻ for bd-type and 2H⁺/e⁻ for bo3-type oxidase) and complex I (2H⁺/e⁻)[58]. ISOVs with complex I variants mainly translocate protons by the quinol oxidases and lack contribution

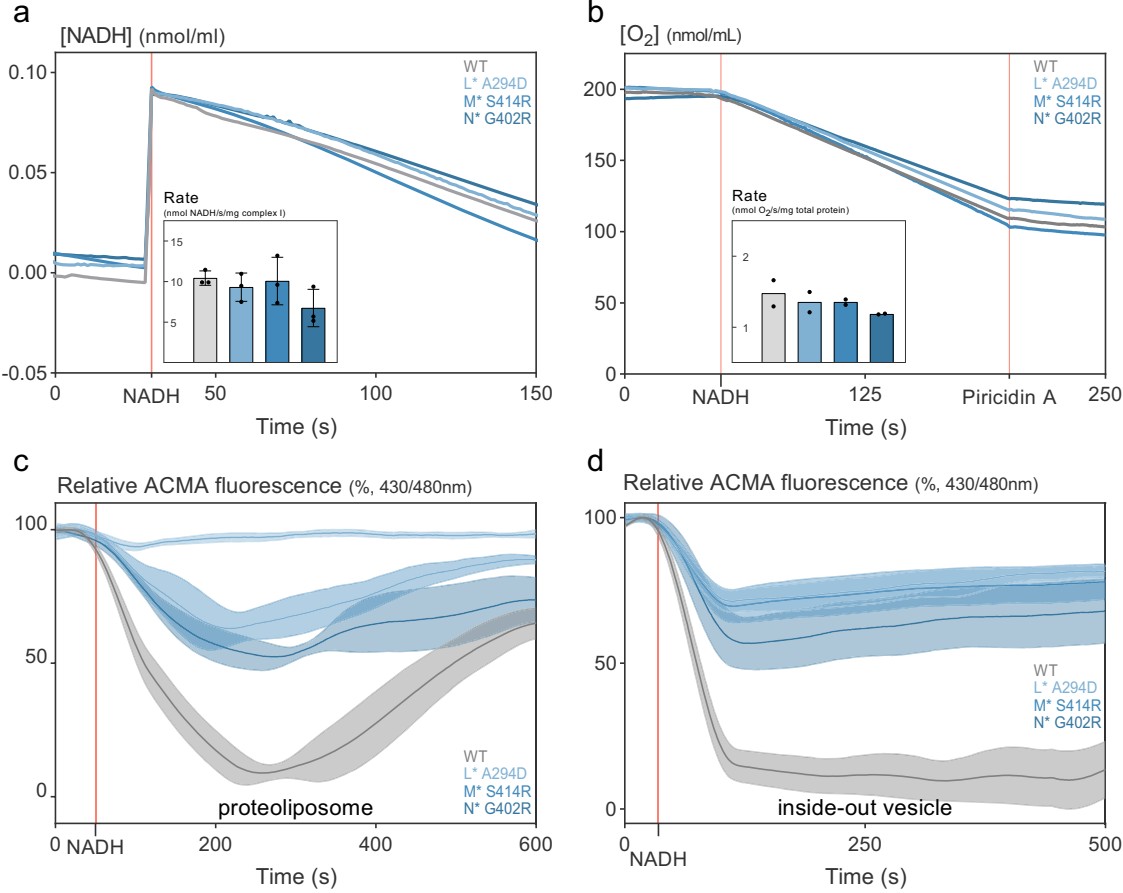

**Fig. 3 High persistence-conferring *nuo*\* variants are impaired in proton translocation. a** Complex I variants show similar NADH/decyl ubiquinone oxidoreductase activity as the wild-type complex. Complexes were reconstituted in liposomes and NADH concentration was monitored at 340 nm (one representative of $n = 3$). Average linear rates of NADH oxidation per mg complex I after NADH addition (inset, means ±stdev, $n = 3$) were not different (one-way ANOVA). Accounting for reconstitution efficiency, orientation, and activity of empty liposomes (see "Methods"), these rates amount to an overall NADH/decyl ubiquinone activity of 2.7 U mg$^{-1}$, which is comparable to literature[154]. **b** The Nuo\* variants and the wild-type complex transport electrons from NADH into the ETC with similar efficiency. Respirometry using a Clark-type electrode with membranes comprising complex I or its variants was used to measure the reduction of O$_2$ by electrons which were released from NADH by complex I (one representative of n=2). Average linear rates of O$_2$ consumption per mg protein after NADH addition (inset, means) were not different (one-way ANOVA) and amount to an overall NADH oxidase activity of 0.16 U mg$^{-1}$, which is comparable to literature[154]. The second red vertical line indicates addition of piericidin A (10 μM), a potent complex I inhibitor, which indicates that O$_2$ reduction indeed originates from NADH oxidation by complex I and that variants and wild-type complex I are equally sensitive to this inhibitor. **c**, **d** Proton translocation is impaired in all variants as estimated from the difference in fluorescence quenching of the pH-sensitive fluorophore ACMA in **c** proteoliposomes containing reconstituted complex I and in **d** ISOVs generated from membranes (means ±sems; $n = 3$). Relative fluorescent values were obtained by comparing the fluorescence to the start and through rescaling between 0 and 100% where 100% is the maximum value of each sample and 0% is the lowest value of the run (i.e. wild-type complex I which showed ±50% quenching, similar to values found in literature[154]). Proteoliposomes are leaky and therefore revert the ACMA quench over time. ISOVs, while additionally showing the activity of the terminal oxidases, are much tighter and show no reversion. In **a**, **d**, reactions were started by adding NADH (first red vertical line) and individual graphs are nudged horizontally and, in **a**, **b** also vertically to allow comparison. See also Supplementary Fig. 3. Source data are provided as a Source Data file.

from complex I (so 1–2 H$^+$/e$^-$), while complex I significantly contributes to proton translocation in wild-type ISOVs (so 3–4 H$^+$/e$^-$), which concurs with the 50–70% lower ACMA quenching in ISOVs with complex I variants compared to wild-type ISOVs (Fig. 3d). Overall, these results demonstrate that the identified *nuo*\* mutations in complex I target proton translocation and uncouple translocation from electron transfer.

**Neither impaired antibiotic uptake nor decreased energy levels fully explains complex I-dependent increased persistence.** Given their compromised proton translocation, we hypothesized that the *nuo*\* mutants could have a decreased proton motive force (PMF). Previously, persisters were shown to tolerate aminoglycosides by means of a decreased PMF leading to lower antibiotic

uptake[59]. To test this hypothesis, we assessed the electrical gradient, i.e. the part of the PMF driving the uptake of charged aminoglycosides, using bis-(1,3-dibarbituric acid)-trimethine oxonol (DiBAC$_4$(3)), a fluorescent dye whose uptake inversely correlates with the electrical potential across the membrane[18]. Using flow cytometry, we assessed the level of DiBAC$_4$(3) of single cells to quantify their electrical gradient. Treatment with the dissipator carbonyl cyanide m-chlorophenyl hydrazone (CCCP) served as a depolarized control. The *nuo*\* mutants showed a decreased electrical potential compared to the wild type (higher curves in Fig. 4a), resembling an expected consequence from the impaired proton translocation in complex I. However, the decrease in electrical gradient is minor—as expected, since the contribution of pH to the electrical gradient is usually only minor[58,60]—and does not agree with the completely depolarized control that received CCCP

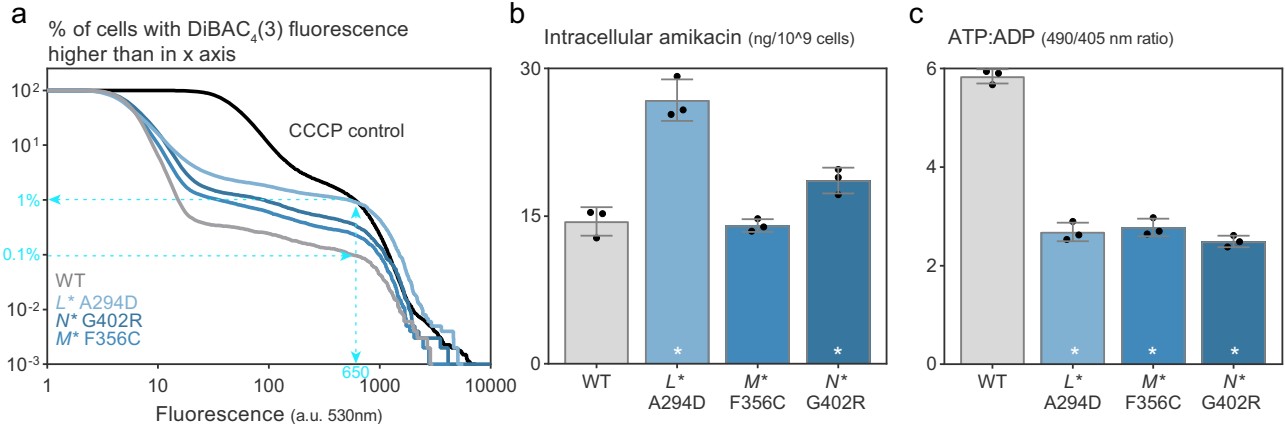

**Fig. 4 High-persistence *nuo\** mutants have no impaired antibiotic uptake and do not solely depend on a decreased ATP:ADP ratio. a** Single-cell fluorescence intensity distributions measured via cytometry where fluorescence corresponds to DiBAC$_4$(3) uptake and serves a proxy for the extend of electrical depolarization. The data were plotted as "1-cumulative distributions", showing the percentage of cells with an intensity above the fluorescence value specified in the *x*-axis to aid comparison between the strains. *nuo\** mutants in stationary phase show a minor decrease in electrical potential, indicated by the higher fraction of cells (higher curves) that have high DiBAC$_4$(3) uptake (starting from a fluorescence value of ±10) (one representative of $n = 2$). Control samples treated with CCCP (in black, average across all repeats and strains), a protonophore and potent inducer of antibiotic tolerance[70], show the expected increase in dye uptake in all cells for a depolarized sample. Cyan lines, arrows and numbers were added to put the results in biological context: the wild-type persister level (0.1% of the population, see Fig. 2e) corresponds to fluorescent level above 650. Only <1% of the *nuo\** mutants have such a high level of DiBAC$_4$(3), whereas these strains carry on average 50% persister cells (Fig. 2d, e, Supplementary Fig. 2f–i). **b** Intracellular amikacin concentrations of stationary phase cultures of *nuo\** mutants measured in a bioassay are similar to the wild type or higher, confirming that antibiotic uptake is not decreased in the high-persistence mutants (means ±stdev, $n = 3$; *$p < 0.01$ for comparison to the wild type from a one-way ANOVA with Dunnett's posttest). **c** In stationary phase (16–20 h after 1/100 dilution of an overnight culture), ATP:ADP ratios are significantly decreased as measured by the ratiometric GFP, Perceval (means ±stdev, $n = 3$; *$p < 0.0001$ for comparison to the wild type from a one-way ANOVA with Dunnett's posttest). However, *nuo\** mutants show similar decreases in ATP:ADP ratios in exponential phase when they do not display increased persistence (Supplementary Figs. 4b, 2h). See also Supplementary Fig. 4. Source data are provided as a Source Data file.

treatment. More importantly, under the assumption that depolarization fully explains the tolerant state, the differences in electrical gradient do not match with the differences in persister levels. For example, the 0.1% persister level of the wild type agrees with an electrical potential that is only present in 1% or less of the cells in the *nuo\** mutants (cyan lines in Fig. 4a) while these mutants have a persister level of ±50% (Fig. 2d, e, Supplementary Fig. 2f–i). Furthermore, and consistent with the small decrease in electrical gradient, *nuo\** mutants did not display decreased antibiotic uptake, as indicated by similar or even higher intracellular amikacin concentrations compared to the wild type (Fig. 4b). In addition, increased tolerance by a decreased antibiotic uptake as a result of a decreased electrical gradient would also not explain why the *nuo\** mutations also offer protection against ofloxacin (Supplementary Fig. 2g), a member of the fluoroquinolone antibiotics whose uptake does not dependent on an electrical gradient[61–63] and which is also readily taken up by the *nuo\** mutants (Supplementary Fig. 4a). Altogether, these experiments show that a difference in antibiotic uptake does not explain the increased persister levels in the *nuo\** mutants.

Another consequence of the decreased proton translocation in the mutants could be a lowered ATP generation by the PMF-driven ATP synthase. Recently, a lower energy status has been linked to persistence[15,21,41–43], while another study did not find decreased ATP levels in persisters[26]. We used a genetically encoded fluorescent reporter, Perceval[64], to measure ATP:ADP ratios in stationary phase populations. The *nuo\** mutations led to a decreased ATP:ADP ratio compared to the wild type (Fig. 4c), which would be in line with the hypothesis of a de-energized cell status as cause of tolerance. However, the ATP:ADP ratios in the *nuo\** mutants are also decreased in the exponential phase (Supplementary Fig. 4e), a condition in which none of the mutants show increased tolerance (Supplementary Fig. 2h). While

lower energy status might still contribute to the increased tolerance as an important prerequisite, we conclude that neither decreased antibiotic uptake nor a lower energy status by altered PMF can single-handedly explain the increased persister levels of the *nuo\** mutants, a result which we also confirmed for the evolved mutants of the uropathogenic strain (Supplementary Fig. 4c–g).

**Increased cytoplasmic acidification in *nuo\** mutants is causal for increased persistence.** Next, we hypothesized that the impaired proton translocation of the complex I variants might influence pH homeostasis during metabolic perturbations, which in turn could lead to increased persistence. To test this hypothesis, we used the ratiometric fluorescent pH sensor pHluorin[65] (Supplementary Fig. 4h) compared the cytoplasmic pH between the wild type and the high-persistence *nuo\** mutants. In exponential phase, a condition where we do not observe increased persistence in *nuo\** mutants (Supplementary Fig. 2h), the pH in the *nuo\** mutants did not significantly differ from the one in the wild type exhibiting an average pH value of 7.73 ±0.01 (Supplementary Fig. 4j). However, in stationary phase, all high-persistence *nuo\** mutants showed a significantly more acidified cytoplasm compared to the wild type (pH$_i$ = 6.79 ±0.04 versus pH$_i$ = 7.39 ±0.05; respectively, $p < 0.0001$) (Fig. 5a, Supplementary Fig. 4i). The increased persistence in the *nuo\** mutants correlates with stronger acidification of the cytoplasm (Supplementary Fig. 4k), likely being elicited by the perturbation in metabolic homeostasis during entry into stationary phase.

To examine whether acidification of the cytoplasm is a general trait of metabolically induced persisters, we measured the cytoplasmic pH dynamically in time during well-controlled carbon nutrient shifts that are known to induce extremely high-persistence levels[38]. During exponential growth on glucose, we

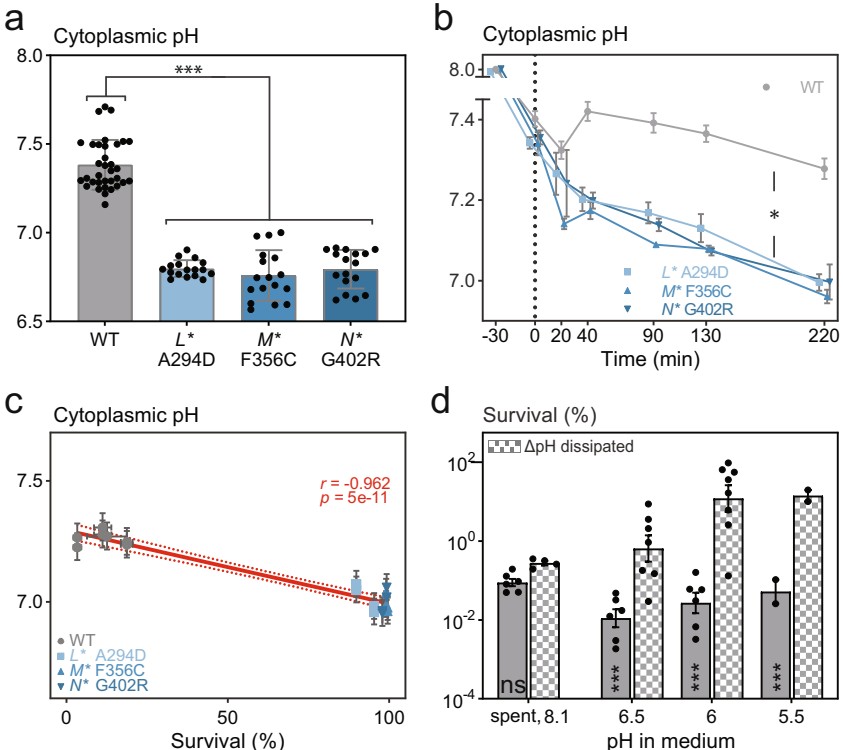

**Fig. 5 Increased cytoplasmic acidification in *nuo\** mutants is causal for increased persistence. a** Cytoplasmic pH measured in cell populations using the ratiometric pHluorin sensor, is significantly decreased in the high-persistence *nuo\** mutants in comparison to the wild type in stationary phase, where the external pH is 8.1 ± 0.06 (means ±stdevs, n≥17; *** $p<0.001$ for a phenotype-level comparison from a linear mixed model). **b** Cytoplasmic pH of cells growing in exponential phase on M9 glucose medium decreases in all strains upon a shift to M9 fumarate medium (dotted line; pH = 7), known to induce persister formation. Dynamic measurement of cytoplasmic pH after a nutrient shift shows continued decreases over time only in the high-persistence *nuo\** mutants (means ±sems, n≥6; * slopes are different based on AIC criterion with the wild-type slope which is non-significantly different from 0 based on an F test). **c** Survival of amikacin treatment (for 4 h with 400 µg ml⁻¹) assessed 30 min after the nutrient shift (in the same experiment as shown in **b** reveals a significant negative correlation between survival and cytoplasmic pH (means ±sems, n≥4). The linear regression line (±95% confidence interval) along with Pearson *r* and *p* values (top right corner) are shown in red. **d** Cytoplasmic acidification is causal for increased persister levels. Resuspending stationary phase cells in sterile spent medium buffered at different pH values increased the survival of amikacin treatment (for 5 h with 400 µg ml⁻¹) from a pH of 6.5 and below in the wild type only when benzoate and methylamine are simultaneously added (ΔpH dissipators at 40 mM; boxed bars) to equilibrate cytoplasmic to extracellular pH. Adding this weak acid-base pair does not influence tolerance levels in unbuffered spent medium that has a pH of 8.1 ±0.06 (means ±stdevs, n ≥ 2; ***$p < 0.001$ and ns = non-significant for within-pH comparison from a linear mixed model with Šidák's posttest). See also Supplementary Figs. 4–5. Source data are provided as a Source Data file.

again found a highly similar intracellular pH of 7.97 ±0.02 in all strains (Fig. 5b), which dropped to 7.35 ±0.03 directly after the shift to fumarate in all strains. While in the wild type, the pH stabilized around this level, the cytoplasm of the high-persistence *nuo\** mutants continued to acidify to pH values around 7.07 ±0.13 (at 220 min after the nutrient shift) (Fig. 5b). Thus, also with a completely different way of perturbing metabolic homeostasis and inducing persistence, we find that the cytoplasm acidifies, and that this acidification is stronger in the *nuo\** mutants. Remarkably, also here, the decreased pH of the *nuo\** mutants correlated with an increased tolerance (Fig. 5c, Supplementary Fig. 5a), generalizing the findings of a lowered pH and increased tolerance to a different perturbation of metabolic homeostasis.

Intrigued by the correlation between the decreased cytoplasmic pH and increased tolerance, both in the stationary phase and upon the sudden shifts between glucose and fumarate, we asked whether the increased cytoplasmic acidification in the *nuo\** mutants is causal for the increased tolerance levels. To test this, we aimed to lower the cytoplasmic pH in the wild type to see whether this also results in increased antibiotic tolerance. To accomplish this, we resuspended stationary-phase cells in spent medium buffered at a range of pH values and added a membrane-

permeable pair of a weak acid and base, potassium benzoate and ethylamine hydrochloride[65], to equilibrate the cytoplasmic pH with the pH of the spent medium. Here, using the wild type, we found that acidification of the cytoplasm increases persistence levels (Fig. 5d, Supplementary Fig. 5b). Thus, the decreased cytoplasmic pH, which seems to be a general feature associated with persistence-inducing conditions involving metabolic perturbations, is further aggravated in the *nuo\** mutants, likely due to their compromised proton translocation, being causal for their increased tolerance.

**RpoS contributes to increased persistence.** Next, we asked how the lowered intracellular pH in the *nuo\** mutants could lead to increased tolerance. In our previous study, we found that the persisters generated through sudden nutrient shifts have an increased RpoS response with higher RpoS levels[26]. The stationary-phase specific nature of the increased persistence in the *nuo\** mutants also points to a role for RpoS, which is the 'stationary phase' sigma factor. As RpoS is posttranscriptionally and posttranslationally upregulated at low pH[66–68], we hypothesized that the increased tolerance could be due to an increased pH-induced RpoS response.

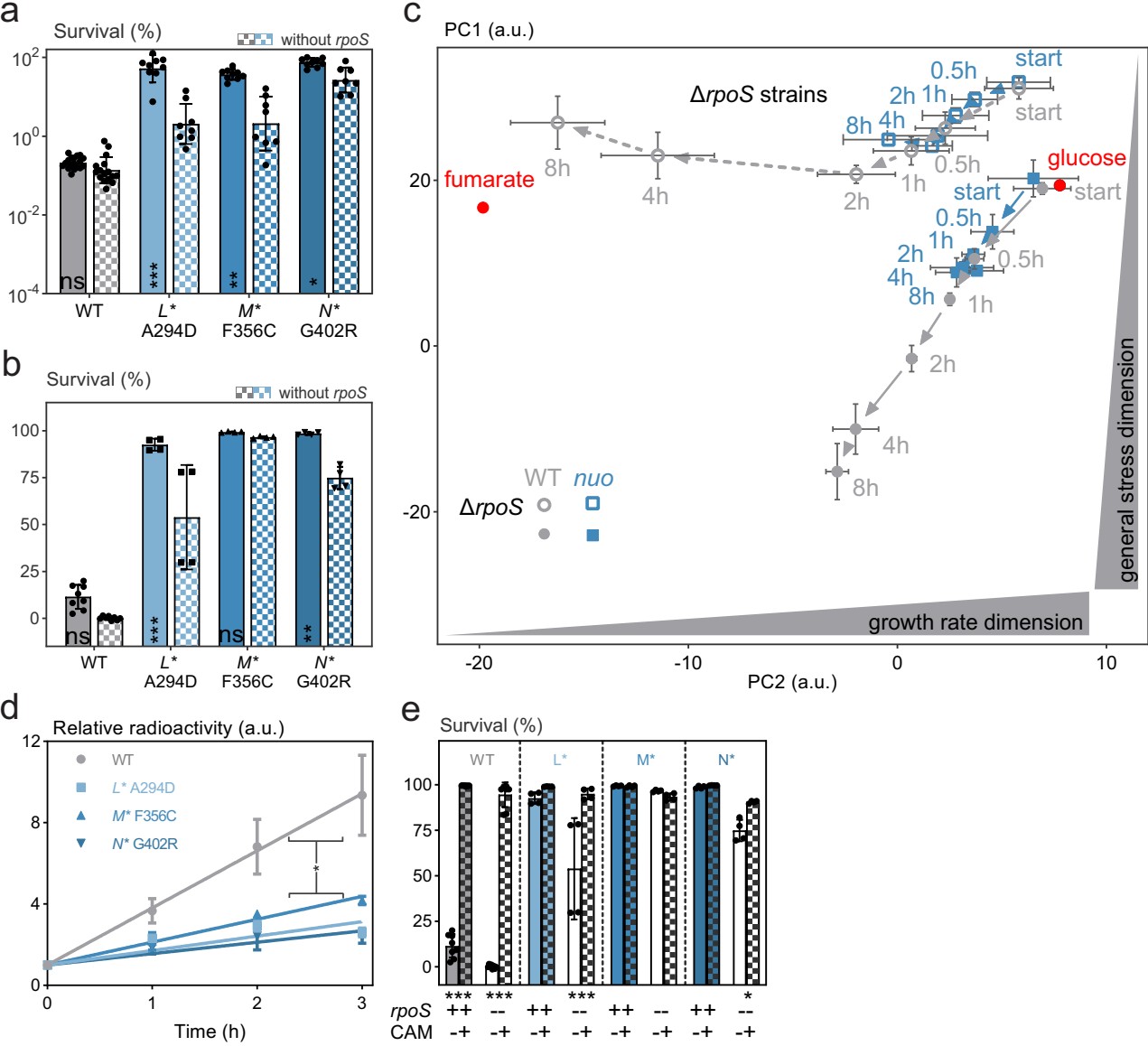

**Fig. 6 RpoS and decreased protein synthesis contribute to increased persistence in *nuo\** mutants. a**, **b** Increased tolerance of the *nuo\** mutants is partially RpoS-dependent in **a** stationary phase (amikacin for 5h with 400 μg ml$^{-1}$) and **b** upon the glucose-to-fumarate shift (amikacin for 4h with 400 μg ml$^{-1}$) (means ±stdevs, n≥4; * $p<0.05$, ** $p<0.01$, *** $p<0.001$ and ns = non-significant for within-strain comparison from on a linear mixed model with Šidák's posttest). **c** Proteome changes after a shift from growth on glucose to fumarate (red dots) indicate that the proteome of high-persistence *nuo\** mutants is in a locked state (means ±stdevs, n≥2). The wild type with functional RpoS (closed gray symbols) shows an activation of the RpoS regulon as it moves down on the stress dimension of the PCA plot over time (see Methods), while removing *rpoS* (open gray symbols) leads to an adaptation to steady-state growth on fumarate and a migration along the growth rate dimension. Starting from 30 min after the shift to fumarate, the proteome of *nuo\** mutants remains unchanged, regardless of the presence of RpoS (in blue; average of *nuoM\** and *nuoN\** mutants). The *nuoL\** mutant was omitted as, in this particular experiment, it did not show strong and persistent acidification and hence, displayed a deviating proteome change (Supplementary Fig. 5f, g), which further substantiates our finding. **d** Translation activity is lower in *nuo\** mutants in stationary phase as measured by the incorporation of radioactive H$^3$-leucine (means ±stdevs, n ≥ 2; * $p < 0.05$ for a phenotype-level comparison from a linear mixed model). **e** Chloramphenicol (CAM) supplementation (25 μg ml$^{-1}$ at the time of the shift) induces tolerance towards amikacin (for 4h with 400 μg ml$^{-1}$; 30 min after the shift) upon shifting from glucose to fumarate in the WT with/without *rpoS* and the *nuoL\** and *nuoN\** mutant lacking *rpoS* (means ±stdevs, n = 4; * $p < 0.05$, *** $p < 0.001$ for within-strain comparison between presence/absence of CAM from a linear mixed model with Šidák's posttest). See also Supplementary Figs. 5, 6. Source data are provided as a Source Data file.

Indeed, supporting a role for RpoS, we found that deleting *rpoS* reduces persistence of the *nuo\** mutants to some extent in stationary phase conditions (Fig. 6a) and also upon the glucose-to-fumarate shift (Fig. 6b). A mechanism that might explain how cytoplasmic acidification leads to increased RpoS levels is via increased ppGpp levels through a pH-based inhibition of both the spontaneous and the SpoT-dependent ppGpp hydrolysis, which

we found to occur at pH values characteristic for persisters (Supplementary Fig. 6; see "Methods"). Surprisingly, however, high-persistence *nuo\** mutants lacking both RelA and SpoT, which are incapable of producing (p)ppGpp, remain highly antibiotic-tolerant in stationary phase conditions (Supplementary Fig. 5c). Thus, alternative and/or redundant routes need to exist that connect pH to RpoS induction. Moreover, since the *nuo\**

mutants lacking RpoS are still more antibiotic tolerant than the wild type (Fig. 6a, b) and since the correlation between cytoplasmic pH (Supplementary Fig. 5d, e) and tolerance remains significant in those strains (Supplementary Figs. 4k, l, 5a), a route additional to RpoS needs to connect the stronger decrease in cytoplasmic pH with the increased antibiotic tolerance.

**Decreased pH inhibits translation and causes increased persistence.** To investigate how the strong pH decrease in the nuo* mutants could cause the increased tolerance, we subjected the wild type and the nuo* mutant to proteome analyses at different time points during the controlled shift from glucose to fumarate, where we had found that the nuo* mutants acidify more strongly than the wild type (Fig. 5b). We mapped the high-dimensional proteome data onto a two-dimensional space created by a previously performed principal component analysis of a similar dataset (Fig. 6c, Supplementary Fig. 5f; see "Methods")[26,69]. One dimension reflects a "growth axis" along which proteomes from cultures growing at steady-state in different growth conditions cluster. The other dimension reflects a "stress axis". It was previously found that the proteome of the wild type after the nutrient shift leading to persister formation, moved down the "stress" dimension, in a process that continued over >8 h[26].

Here, agreeing with these previous results, we find that wild-type cells, when shifted from glucose to fumarate, display proteome dynamics moving down the stress dimension (closed, gray symbols in Fig. 6c), characterized by induction of the global stress response[26]. Secondly, we found that the proteome of the wild-type strain lacking rpoS (open, gray symbols in Fig. 6c) changes steadily over time towards a proteome of cells with normal steady-state growth on fumarate (red dot in Fig. 6c). This is in line with the previous finding that, in an ΔrpoS mutant, almost all cells resume growth after a rapid shift to fumarate[26]. Furthermore, this observation supports our earlier conjectured hypothesis[32] that an ΔrpoS mutant strain invests resources in restoring metabolic homeostasis, and not in massive activation of stress responses, which is also corroborated by a somewhat restored pH homeostasis (i.e., reversal of the initial acidification) in the absence of RpoS in the wild type after a shift to fumarate (Supplementary Fig. 5d).

The proteome of nuo* mutants carrying a functional rpoS gene (closed, blue symbols in Fig. 6c) showed changes in the direction of an induced stress response for the first 30 min after the nutrient shift, similar to the changes of the wild-type proteome. Surprisingly, however, after 30 min the proteome stops changing abruptly (Fig. 6c), suggesting that the proteome might be locked in a frozen state with no further net turnover of proteins (i.e., either no or a balanced translation and degradation). Notably, it was also around 30 min that the cytoplasmic pH started to deviate between the wild type and the nuo* mutants (Fig. 5b) and that a correlation emerged between cytoplasmic pH and antibiotic tolerance (Fig. 5c, Supplementary Fig. 5a). Additionally, nuo* mutants lacking rpoS show a similar halt in proteome dynamics (Fig. 6c), which is in agreement with the only minor decreases in tolerance upon deletion of rpoS in these backgrounds. Thus, these data suggest that after about 30 min, which is when the pH in the nuo* mutants has dropped to pH values below 7.2, net protein turnover comes to a halt.

To test whether protein turnover is also halted in the stationary phase conditions on complex medium, we performed radioactive labeling experiments to follow protein synthesis. Here, we found that translation is indeed impaired in all nuo* mutants under the conditions that they were originally selected for (Fig. 6d), excluding the possibility that any net changes in the proteome were masked by a translation activity that balances out protein

degradation. Thus, we conclude that the strong cytoplasmic acidification in the nuo* mutants halts changes in proteome turnover. We conjectured that halted protein translation, because of inactivity of amikacin's antibiotic targets, could render the nuo* mutants more tolerant than the wild type.

To test whether the halted translation in the nuo* mutants is indeed responsible for increased persistence, we artificially blocked translation after shifting cells to fumarate by the addition of a sublethal dose of chloramphenicol and checked for tolerance against amikacin after 30 min. Here, we found that such a pretreatment, similar to what was found previously[70], results in a strong induction of persistence in the wild type, comparable to the levels in the nuo* mutants without pretreatment (Fig. 6e). Additionally, such an artificial inhibition of translation also abolishes the need for rpoS both in the wild type and nuo* mutants. Thus, we conclude that the pH-induced halted translation in the nuo* mutants is indeed the main factor responsible for the observed increased persistence against amikacin, bypassing the need for a full-blown activation of stress responses by RpoS as mechanism for tolerance as is the case for the wild type.

## Discussion

Here, we present a mechanistic model, which explains how persister cells are formed upon perturbations of metabolic homeostasis (Fig. 7). In this model, strong metabolic perturbations, such as the shift into stationary phase or abrupt nutrient shifts, lead to cytoplasmic acidification, likely through protons released through continued ATP consumption without ATP regeneration, which has been linked to acidification in other systems[71–75]. Mutations in complex I that compromise proton translocation result in a further decreased cytoplasmic pH upon metabolic perturbations. Part of the following induction of persistence, especially in the wild type, is conferred by the RpoS-dependent global stress response. In the nuo* mutants with increased cytoplasmic acidification, persistence is additionally conferred by a global halting of protein synthesis. At the system level, additional feedback and interaction mechanisms exist: an RpoS response inhibits restoration of metabolic homeostasis[26] and the pH-induced halted protein synthesis further prevents restoration of homeostasis, while also halting a further activation of the RpoS response. Our newly identified mechanism also agrees with a growing body of recent research that suggest that bacterial metabolism, chemiosmotic homeostasis and membrane bioenergetics—in addition to drug uptake and activity of primary targets—dictates the lethality of bactericidal antibiotics[76–78].

Our work identifies complex I as an important persistence factor. Loss-of-function mutations in complex I (e.g., transposon or frameshift mutations) are long known to abrogate the growth advantage in long-term stationary-phase conditions of rpoS 'GASP' mutants in absence of antibiotics[79]. More recently, such mutations were shown to increase gentamicin tolerance in E. coli[80] and aminoglycoside resistance in E. coli[81–83] and Pseudomonas aeruginosa[84] through a reduced PMF and antibiotic uptake. Such an effect could also be mimicked by artificially decreasing complex I activity[85–88] and was recently proposed to underlie the antibiotic tolerance of a point mutant in nuoN in P. aeruginosa[89]. Here, we describe a new way in which complex I alters antibiotic susceptibility, i.e. by changes that solely impair proton translocation activity of the complex. Mutations identified in this work specifically lie in the proximity of the three previously confirmed proton pathways in NuoL, M and N and were found at the interface of the NuoAHJK subunits, thereby corroborating a previously hypothesized fourth channel at this site[51,53,90]. Additionally, the nuo* mutants still take up antibiotics

**Fig. 7 Model connecting metabolic homeostasis, cytoplasmic acidification and two paths for persister cell formation.** A strong perturbation of metabolic homeostasis by nutrient shifts or entry in stationary phase acidifies the cytoplasm. Such acidification could occur due to an imbalance between ATP consumption and regeneration. In these conditions, complex I acts as a regulator of cytoplasmic pH: wild-type proton translocation activity counter-balances acidification; impaired proton translocation in the nuo* mutants enhances cytoplasmic acidification after ±30 min of perturbation of metabolic homeostasis. Mild cytoplasmic acidification leads to increased persistence by an increased RpoS response, possibly mediated by inhibition of SpoT-mediated hydrolysis of (p)ppGpp (dominant path in black arrows in the wild type). Stronger cytoplasmic acidification halts protein synthesis and renders antibiotic targets inactive leading to a second persistence mechanism (dominant path in red arrows in nuo* mutants with impaired complex I function.) At various points in this model, feedback loops are in place that affect persister formation. For instance, RpoS inhibits adaptation to maintain metabolic homeostasis, recovering from the initial pH drop. In turn, strong intracellular acidification via halted protein synthesis inhibits full RpoS activation.

and show cross-tolerance towards other antibiotics, in contrast to the collateral hypersensitivity towards other antibiotics of (loss-of-function) mutations in nuo (and other genes) described in the context of aminoglycoside resistance due to reduced drug uptake[83,91]. Therefore, the nuo* mutations identified in the current study constitute a new mechanism in which complex I influences antibiotic survival in bacteria, i.e. by inducing persister formation through increased cytoplasmic acidification upon strong perturbations of metabolic homeostasis.

In this work, we uncovered that cytoplasmic acidification—induced by perturbations of metabolic homeostasis, i.e. upon nutrient shifts or entry into stationary phase—acts as a signal for persister formation in E. coli. In fact, a drop in cytoplasmic pH was observed in Salmonella upon macrophage engulfment and this acidification was furthermore important for the injection of modulating effector proteins that allow Salmonella to survive inside the vacuole[92]. At the same time, but without a connection to cytoplasmic acidification, Helaine and colleagues showed that macrophage internalization induced intracellular persister formation[93]. Indole, the inter-kingdom signal molecule that was previously connected to the persister state[94–96], was shown to regulate intracellular acidification upon entry in stationary phase[97] although very recently, the tryptophanase that catalyzes the formation of indole, not indole itself, was found to cause a lower pH in persisters[98]. In Mycobacterium smegmatis, bactericidal activity of antibiotics was linked to cytoplasmic alkalization during treatment[99] while cytoplasmic acidification was recently found important for fully fledged resistance in marR E. coli mutants[100]. Beyond persisters and antibiotic survival, a drop in the cellular pH is more commonly associated with metabolic perturbations and dormancy. For instance, the pH of spores of Bacillus megaterium and Bacillus cereus is 1 to 1.3 pH units lower than the pH of growing cells, but rises rapidly upon germination[101–103]. In yeast, cytoplasmic pH was identified as a second messenger for glucose levels where low intracellular glucose induces acidification[104]. Even beyond microbial life, a pH drop seems to be a dormancy signal, as intracellular pH is a dormancy marker in strawberry plants[105], and has been shown to regulate dormancy in shrimp embryos[106]. Thus, across different species and even kingdoms, cytoplasmic acidification might act as

a 'low-level' signal, indicating strong perturbations of metabolic homeostasis, ultimately causing cells to enter 'safe states' such as dormancy or persistence.

Since RpoS is known to be upregulated at low pH[66–68] its role in persistence after cytoplasmic acidification was somewhat expected. How cytoplasmic acidification influences the here uncovered halting of protein turnover as they occur in the nuo* mutants after 30 min (Fig. 5d), is less clear as mechanisms to downregulate protein synthesis during growth arrest are not fully understood[107]. The sudden and simultaneous stop of both protein production and degradation, as we inferred from both the sudden stop in proteome changes and the decreased translation activity (Fig. 6c, d), could result from cytoplasmic acidification leading to—as shown in yeast—a glass-like state with decreased intracellular dynamics that induces dormancy[75], similar to the glassy cytoplasm connected with dormancy in other species[108–111]. Additional evidence for the significance of cytoplasm fluidity is the recently identified correlation of protein aggregate formation with persistence[41,112–115], while, at the same time, widespread and pH-dependent aggregation of native-like proteins underlies the transition of the cytoplasm to a solid-like state[75]. Recently, mutations in translation-related genes were selected during laboratory evolution towards increased persistence and shown to affect translation more specifically[116]. Similarly, changes in cytoplasmic pH could also modulate ribosome activity more directly than via the induction of a global glass-like dormant cytoplasm, for example by unbalancing the ionic fluxes of other cations like $K^+$, $Ca^{2+}$, and $Mg^2$, that were recently associated with ribosome functioning and/or antibiotic killing[76,78,117]. Therefore, a decrease in cytoplasmic pH could lead via a myriad of ways (e.g., cytoplasmic rigidity, ionic imbalances) to halted protein turnover and thus to increased tolerance.

With this work, we have unraveled the molecular basis of how metabolically induced persister cells are formed, where the dynamic nature of cytoplasmic pH, as a signal for strong metabolic perturbations, resembles a global connector of metabolism with persister formation. The generic nature of this mechanism, converging at cytoplasmic acidification, is consistent with the fact that numerous effects, including perturbations of many metabolic genes[118,119], abrupt nutrient shifts, changes in extracellular pH[93],

sublethal concentrations of antibiotics and oxidative stress[120] have all been observed to induce tolerance and could all converge in intracellular acidification.

## Methods

**Bacterial strains, media, and cultivation**. All bacterial strains used in this study are listed in Supplementary Data 3, with the most important strains being SX43 or SX2513 (the ancestral lab strains used in most of our experiments as "wild type" or WT, which are close relatives to BW25113), and UTI89 (a uropathogenic *E. coli* strain). Genomic mutants were constructed with either P1*vir* phage transduction as described before[121] or through homologous recombineering described before[122]. Where possible, mutants from an ordered gene knockout library in *E. coli*, the Keio collection[123] were used as template to create new mutants and they were cured from their Km^R-cassette by transformation with pCP20 expressing the FLP recombinase. At all steps, validity of mutants was checked by either PCR with gel electrophoresis or targeted Sanger sequencing or both using primers listed in Supplementary Data 4.

Experiments were performed using M9 minimal medium prepared as previously described[38], Mueller Hinton Broth (MHB, Becton Dickinson; widely used in antibiotic sensitivity testing), lysogeny broth (LB) medium (10 g L^−1 NaCl, 10 g L^−1 trypton, 5 g L^−1 yeast extract with/without 15 g L^−1 agar), M63 minimum salts medium as previously described[65] or 'spent' MHB medium. The carbon source in MHB and LB is an undefined mixture of peptides with minor traces of sugars. The spent MHB medium was prepared by removing wild-type cells from an overnight culture (37 °C, 200 rpm or 1*g*) by centrifugation and filtration through a 0.2 μm filter. For spent MHB medium buffered at a certain pH, the appropriate Good's buffering agent[124] was added (50 mM 2-(N-morpholino)ethanesulfonic acid for pH 5.5 and 6 and 50 mM piperazine-N,N′-bis(2-ethanesulfonic acid) for pH 6.5) with/without potassium benzoate (40 mM) and methylamine hydrochloride (40 mM) to dissipate ΔpH. Next, the pH was adjusted using HCl or KOH and the medium was filter sterilized. For minimal media, carbon source stock solutions were prepared in demineralized water, adjusted to pH 7.0 using NaOH or HCl, respectively, and sterile filtered through a 0.2 μm polyethersulfone (PES) filter. Glucose was added in a final concentration of 5 g L^−1 and fumarate in a final concentration of 2 g L^−1, respectively. Antibiotics, IPTG and arabinose were all prepared in demineralized water and sterile filtered through a 0.2 μm PES filter. Bacterial cultures were cultivated either in 50 ml medium in a 500 ml flask or 10 ml medium in a 100 ml Erlenmeyer flask closed with a 38 mm silicone sponge closure (Bellco Glass) at 37 °C, 300 rpm and 5 cm throw (5 × *g*) or in 100 ml medium in a 250 ml Erlenmeyer flask closed using cellulose stoppers (VWR) at 37 °C, 200 rpm and 2.5 cm throw (1 × *g*). Overnight cultures were diluted 1:100 in fresh medium and further diluted 1:100 as soon as OD_600 of 0.5 is reached to keep cells in the exponential growth phase.

**Assessment of the antibiotic sensitivity**. MICs, survival levels, and killing curves were determined according to the methods described previously[8,125,126]. For MIC determination an overnight culture was diluted in fresh MHB to an inoculum of $1 \times 10^6$ colony-forming units per milliliter (CFU ml^−1) and incubated in a range of two-fold antibiotic dilutions for 16–20 h. After incubation, the lowest antibiotic concentration where no growth was observed using the absorbance at 595 nm (BioTek), we defined as the MIC. Alternatively, commercially available MIC test strips were used (Liofilchem).

Antibiotic survival level in the stationary phase is obtained by taking the ratio CFU ml^−1 after treatment (usually for 5h) and the total CFU ml^−1 before antibiotic treatment (generally $1–5 \times 10^9$ CFU ml^−1). CFU ml^−1 were obtained by making serial dilutions, spiral plating, and semi-automatic quantification after 2 days of incubation at 37 °C (EddyJet and Flash & Grow). When using spent medium, stationary-phase cells were resuspended in an equal volume of the spent medium and treatment was started immediately.

The antibiotics that were used are all aminoglycosides or fluoroquinolones. We used these two classes as they are potent bactericidal drugs that furthermore can kill sensitive stationary phase cells and result in a biphasic killing pattern, indicative for the survival of persister cells. Furthermore, these antibiotics were used in our previous work which forms the basis of the current work[8].

**Assessment of antibiotic tolerance through cytometry upon nutrient shifts in M9 medium**. Tolerance experiments upon nutrient shifts were always combined with ratiometric pH measurements. Cells were grown in M9 glucose medium containing ampicillin (100 μg ml^−1) and arabinose (20 g L^−1) with regular dilutions to maintain growth in the exponential phase. At OD_600 0.5, pH was measured and persisters were induced by a switch to M9 fumarate medium. At multiple time points after the nutrient switch, samples were taken, and intracellular pH was determined as previously described. For determination of antibiotic tolerance, we treated the cells with amikacin (4 h at 400 μg ml^−1) at 0.5 h after the switch. As a control we blocked translation using a low dose of chloramphenicol (25 μg ml^−1) added direct after the switch to fumarate. After 4 h, amikacin treated cells were diluted 100-fold in LB and their ability to regrow was tracked using flow cytometry (for 3h, see refs. [26,127]). Tolerant cells resumed growth, became bigger and lost their pHluorin fluorescence intensity whereas non-growing cells retained their size and

fluorescent intensity. The fraction of cells that did not resumed growth was determined by observing no cell size changes and no loss of fluorescence. Cells were measured every 30 min from 0 to 3 h after the transfer to LB medium. Fractions of non-recovering cells from the period of 2.5 and 3 h after transfer to LB were used to calculate the fractions of tolerant cells by subtracting the nondividing cells from the total cells added to the culture at the start.

**Evolution experiments**. Experimental evolution using the uropathogenic *E. coli* strain UTI89 was performed as before[8,48]. Briefly, parallel cultures were grown overnight to stationary phase in MHB (±18 h), treated for 5 h with amikacin (400 μg ml^−1; 100–200-fold MIC) to eliminate all non-persisters, washed three times in MgSO_4 (10 mM) to remove antibiotics and diluted 1/100 into fresh MHB to allow another cycle of batch growth. A sigmoidal fit expected for the spread of a mutant in a haploid population was fitted to the data obtained under daily treatment:

$$Y = z - (p_0 \times (z - Z_a))/\left(p_0 + (1 - p_0) \times \left(\frac{1}{1+s}\right)^x\right)$$

with *s* the selective advantage of the assumed mutant and *z* and $Z_a$ are the log-transformed persister fraction of the ancestor and mutant. A horizontal line was fitted to the control data without treatment and compared to a straight line with an extra sum of squares F test to confirm the absence of change of phenotype. All fitting was done in GraphPad Prism 8 using the least squares method.

**Genome-wide next-generation sequencing**. We detect mutations arising in evolved populations or clones by genome-wide sequencing using Illumina's HiSeq platform. Genomic DNA was extracted by using the DNeasy Blood and Tissue Kit (Qiagen). Purity and concentration were assessed by Nanodrop, gel electrophoresis and Qubit. Average insert size of the prepared libraries was ±300 bp and sequencing was conducted at EMBL's Genecore in Germany, the VIB Nucleomics core in Belgium, Eurofins, or the Genomics core of UZ Gasthuisberg in Belgium. Raw data files have been deposited in the NCBI SRA database with following accession IDs: PRJNA498891 (populations of the lab strain), PRJNA270307 (clonal data of the lab strain), PRJNA498717 (uropathogenic populations), PRJNA498708 (uropathogenic clones) and PRJNA768774 (to check the ΔrelAΔspoT deletion strains). The 100–150 bp paired-end output was analyzed using Qiagen's CLC Genomics Workbench version 11.0. Full details on the used workflow and parameters are available upon request. Briefly, reads were mapped to reference sequences after quality control, read trimming and filtering (NC_000913.3 for lab strain, NC_007946.1 and NC_007941.1 for the uropathogen UTI89). Mutation lists were obtained with CLC's low frequency variant detection tool. Further filtering of these lists was based on several quality features and prior experiences (scripts available upon request). Importantly, for clones, cutoff frequencies of 75% were used, while for population analysis, an initial cutoff frequency of 5% was used while later on, this cutoff was dropped for the regions of the target genes specifically and data were filtered on minimum coverage only (coverage >30, forward & reverse read count >0; script available upon request). Clonal results were furthermore confirmed using targeted Sanger sequencing with primers in Supplementary Data 4. While we did not specifically remove intergenic or synonymous mutations, only 3 and 0, respectively, were identified in the entire dataset at or above the cutoff frequency of 5%. The perl-based software package Circos was used (http://circos.ca/) to visualize data. Scripts and configuration files are available upon request. Mutations from the UTI background were added to figures, tables and data based on a pairwise alignment to the lab strain background.

For all the Chi² tests to examine for significant enrichment, we used 200,000 replicates in Monte Carlo simulations to compute p-values and computed the null hypothesis as the condition in which mutations would be hitting randomly in the genome (Supplementary Fig. 1f), in genes (Supplementary Fig. 1g), in genes coding for membrane proteins (Supplementary Fig. 1h), in genes coding for inner-membrane proteins (Supplementary Fig. 1i), in genes of the *nuo* operon (Fig. 2b), in *nuo* genes coding for the membrane-spanning subunit (Fig. 2c, Supplementary Fig. 3a, b). To do so, we used the genome-wide annotations in NC_000913: 4641652bp in total, 4089513bp of which are part of coding regions, 1040006bp of which code for membrane proteins, 822021bp of which code for inner-membrane proteins, 14673bp of which code for complex I of which 7110bp are for genes of the membrane subunit. For the membrane subunits, amino acids were scored to be part of either transmembrane helices or non-membrane-spanning loops based on predictions by Protter (wlab.ethz.ch/protter)[128] and scored to be either hydrophobic or hydrophilic based on a scale computed before[129] and a cutoff value of 0.6.

**Genomically repairing *nuo*-mutation**. P1*vir* phage transduction was used to genomically revert the identified mutations in *nuoL*, *M* and *N* by using the *yfbP*::Km^R Keio mutant as donor strain as described elsewhere[8,121].

**Generating 3D structure of complex I from *E. coli***. No full protein structure exists for complex I of *E. coli*. To this end, we modeled the protein structure of the cytoplasmic domain (nuoBCDEFGI; uniprot IDs: P0AFC7, P33599, P0AFD1, P31979, P33602, P0AFD6) and membrane arm (nuoAHJKLMN; uniprot IDs:

P0AFC3, P0AFD4, P0AFE0, P0AFE4, P33607, P0AFE8, P0AFF0) using the online SWISS modeling server (https://swissmodel.expasy.org/)[130] with 6g2j and 4he8 PDB entries, respectively, as templates (Global Model Quality Estimation of, respectively, 0.56 and 0.67 with 35.8% and 38.5% sequence identity). Next, an overall structure of complex I of *E. coli* was obtained by aligning the modeled structures of the cytoplasmic and membrane domains to the structure of complex I of *Thermus thermophilus* (PDB entry 4hea)[51]. To this end, chains G and L were used as anchors by aligning them through the structural alignment of multiple proteins (STAMP) tool[131] in Visual Molecular Dynamics (VMD)[132] with standard settings. Images were generated by the rendering capabilities of PyMOL[133].

**Purification of complex I and its variants.** Complex I transcribed from the pBAD*nuo nuoF*$_{His6}$ was expressed in, and purified from an *E. coli* strain lacking complex I and the alternative NADH dehydrogenase as performed before by the Friedrich group[55]. Site-specific mutagenesis through PCR with Q5 polymerase (NEB) on this construct using primers nuoL_FW and nuoL_RV, nuoM_FW and nuoM RV and nuoN_FW and nuoN_RV (Supplementary Data 4) generated the variants carrying the point mutations of interest. Specifically, after 1:100 dilution in baffled flasks containing autoinduction medium, BW25113*ΔnuoΔndh* with pBAD*nuo nuoF*$_{His6}$ was grown aerobically until late exponential phase (OD$_{600}$ = 3–4). From that point onwards, purification steps were carried out at 4 °C. Cells were harvested through centrifugation at 3000*g* for 15 min. Cell sediments were suspended with a Teflon-in-glass homogenizer in a 5-fold volume of buffer A (50 mM NaCl, 50 mM MES/NaOH, pH 6.0) with 0.1 mM phenylmethylsulfonyl fluoride (PMSF) and few grains of Dnase I. Cells were disrupted by 4 passes through an Avestin Emulsiflex at 110 MPa and cell debris was removed by low speed centrifugation at 9500*g* for 20 min. Next, membranes were separated from the supernatant by centrifugation at 257,000*g* for 1 h and the membrane sediment was suspended in an equal volume of buffer A* (buffer A + 5 mM MgCl$_2$) with 0.1 mM PMSF. Membrane suspensions were either directly used or stored as sediments at −80 °C.

*n*-Dodecyl-*β*-D-maltopyranoside (DDM) was slowly added during 15 min to the membrane suspension until a final concentration of 2%. After gentle stirring for 30 min, insoluble material was removed at 250,000*g* for 15 min. Anion exchange chromatography on a Fractogel EMD TMAE Hicap (Merck) separated membrane proteins which were washed in buffer A* + 0.1% DDM and eluted using buffer B* (Buffer A* with 350 mM NaCl and 0.1% DDM). Fractions with NADH/ferricyanide oxidoreductase activity were pooled, corrected to contain 20 mM imidazole and applied on Ni-IDA material (Invitrogen) for affinity chromatography. After washing with binding buffer (500 mM NaCl, 50 mM MES/NaOH, 0.1% DDM and 20 mM imidazole (pH 6.3), fractions were eluted with 500 mM imidazole. Fractions with NADH/ferricyanide oxidoreductase activity were pooled, washed three times with buffer A* with 0.1% DDM, concentrated using ultrafiltration (Amicon Ultra-15, Millipore, 100 kDa MWCO) and stored in aliquots at −80 °C.

**Protein concentration determination.** Protein concentration was determined based on the biuret method using BSA as standard[134]. Concentration of purified complex I variants was measured by spectrometry (TIDAS II, J&M Analytik) subtracting the absorbance at 310 nm from that at 280 nm and with an ε of 763 mM$^{-1}$ cm$^{-1}$.

**NADH/ferricyanide oxidoreductase activity.** NADH/ferricyanide oxidoreductase activity of membranes, fractions or purified complex I was determined by following the decrease in absorbance of ferricyanide at 410 nm in buffer A with 0.2 mM NADH using an Ultrospec 1100 *pro* spectrophotometer (Pharmacia)[56] and based on an ε of 1 mM$^{-1}$ cm$^{-1}$.

**O$_2$ reduction assay.** Reduction of O$_2$ by electrons released from NADH by complex I was determined in buffer A* on 5 μl of membranes with a Clark-type oxygen electrode at 30 °C (Hansatech). The reaction was started by adding 1.25 mM NADH while 10 μM piericidin A was used to inhibit the reaction.

**Determination of purity and stability of complex I preparations.** Initial production and stability of complex I were verified using polyacrylamide gel electrophoresis (PAGE). Under denaturing conditions, a sodium dodecyl sulfate (SDS) PAGE was run as described previously[56] with a 3.9% stacking gel and a 10% separating gel. In addition, colorless native (CN) PAGE was performed as described previously[135] with a 3.5% stacking and a 4–13% gradient separating gel (pH 6.0). For nitroblue tetrazolium (NBT) staining the gel was incubated for 5 min with 1 mg ml$^{-1}$ NBT in 100 mM MOPS, pH 8 and the reaction was started by an addition of 100 μM NADH.

In addition to PAGE, stability of purified complex I was verified with thermal shift assays. ThermoFAD measures the intrinsic fluorescence of flavin, a cofactor of complex I, to determine the unfolding temperature of the complex[136]. Briefly, 25 μL of complex I (in buffer A, 1 μg μL$^{-1}$) was heated in a CFX96 qPCR thermocycler (Bio-Rad) with fluorescence measurements at regular interval (ex. 470–500 nm, em. 523–543 nm). After subtracting blank data, a first derivative was calculated to obtain melting points as inflection points/maxima peaks. In addition, a Thermofluor-adapted assay[137] made use of the increased fluorescent yield

reaction of 7-diethylamino-3-(4′-maleimidylphenyl)-4-methylcoumarin (CPM) upon reaction with thiol groups that are released upon heating and unfolding. 300 pmol of complex I cysteines (1 μg μL$^{-1}$ of complex I in buffer A) was mixed with a 5-fold excess of CPM, overlaid with silicone oil and assessed in a Perking Elmer LS-55 fluorescence spectrometer (ex. 384 nm, em. 470 nm). A dose-response equation was fitted to the data in GraphPad Prism 8 using the least squares method:

$$Y = \text{Bottom} + (\text{Top} - \text{Bottom})/\left(1 + 10^{((\text{LogEC}_{50}-X)*\text{HillSlope})}\right))$$

A first derivative was calculated from this fit to obtain melting points as inflection points/maxima peaks[57].

**Reconstitution of complex I into liposomes.** Purified complex I was reconstituted in liposomes at 4 °C[138]. Briefly, complex I (2 mg ml$^{-1}$) was mixed with a 4-fold excess of *E. coli* polar lipids (w/w; extract from Avanti) which were dissolved in lipid-buffer (5 mM MES/NaOH, pH 6.0) and DDM (20 mg ml$^{-1}$) by sonication. The mixture was stirred gently for 10 min before BioBeads SM-2 (Bio-Rad) were added in an 8-fold excess, accounted for their binding capacity of and the presence of DDM in the sample and stirred gently for another 3 h. The proteoliposomes were sedimented by centrifugation for 45 min at 150,000*g*, resuspended in proteoliposome buffer (5 mM MES/NaOH, pH 6.0, 50 mM NaCl, 5 mM MgCl$_2$) by gentle pipetting, extruded with 31 passes through a 0.1 μm polycarbonate membrane and used on the same day. The NADH/ferricyanide oxidoreductase activity before and after addition of 0.5% DDM showed no significant differences between variants and wild type. The ratio of both activities was with on average 0.75, indicating that 75% of the complex was oriented in the liposomes so that the NADH binding site was accessible from the buffer. Comparing to the NADH/FeCN oxidoreductase activity of purified proteins, the average protein reconstitution efficiency was 21%.

**Preparation of ISOV vesicles.** BW25113*ΔnuoΔndh* cells with pBAD*nuo nuoF*$_{His6}$ were induced and harvested as described above. Inside-out vesicles (ISOVs) containing the complex I variants and endogenous lipids and other proteins; were prepared at 4 °C from frozen cells following a slightly modified, previously described procedure[139]. Cell sediments were suspended in equal amount of washing buffer (50 mM KH$_2$PO$_4$/KOH, 5 mM MgSO$_4$, pH 7.5), sedimented (10 min, 4,500 *g*), suspended 6:1 in lysis buffer (washing buffer with 1 mM dithiothreitol, 0.1 mM PMSF and some grains of Dnase I) and disrupted by a single pass through a 40 ml French Pressure Cell at 55 MPa (SLM-Aminco). Centrifugation removed debris (20 min at 9500*g*). ISOVs were separated from the supernatant by two ultracentrifugation steps (70 min and 15 min at 257,000*g*) and resuspended in washing buffer. ISOV vesicles were kept on 4 °C and used on the same day.

**Proton translocation and electron transfer activity.** Proton translocation activity of complex I, either reconstituted in liposomes or in ISOVs, was measured by monitory quenching of fluorescence of 9-amino-6-chloro-2-methoxyacridine (ACMA)[138]. Briefly, proteoliposomes were incubated at 30 °C for 1 min with 0.2 μM ACMA and 60 μM decyl-ubiquinone in ACMA buffer (5 mM MES/NaOH, 50 mM KCl and 2 mM MgCl$_2$ pH 6.0). The reaction was started by adding 100 μM NADH and ACMA fluorescence was followed on a Perking Elmer LS-55 fluorescence spectrometer through time (ex. 430 nm, em. 480 nm). For measurements of proton transport using ISOVs, no decyl-ubiquinone was added. Similarly, without ACMA, NADH/decyl ubiquinone oxidoreduction was followed by monitoring the decreasing NADH concentration by spectrometry at 340 nm with an ε of 6.3 mM$^{-1}$ cm$^{-1}$ (TIDAS II, J&M). As a control, the NADH/decyl ubiquinone oxidoreductase activity of empty liposomes was measured to be 15% of the activity of complex I-containing proteoliposomes.

**Measurement of the electrical gradient in vivo.** The electrical gradient across the membrane was determined using electrical-gradient dependent uptake of bis-(1,3-dibarbituric acid)-trimethine oxonol (DiBAC$_4$(3))[8]. Stationary phase cultures were incubated for 10–20 min with 10 μg ml$^{-1}$ DiBAC$_4$(3). Single-cell uptake was measured on a BD influx cytometer (ex. 488 nm, em. 530/40 nm; 100,000 cells) and analyzed using FlowJo v10.3 (FlowJo, LLC). As control, samples were incubated for 1 h with 500 μM of the protonophore carbonyl cyanide m-chlorophenyl hydrazone (CCCP) prior to staining with DiBAC$_4$(3).

**Intracellular antibiotic accumulation measurement.** To measure intracellular uptake of antibiotics, 50 ml stationary phase cultures were incubated with amikacin (100 μg ml$^{-1}$) or ofloxacin (5 μg ml$^{-1}$) for 60 min. Cells were separated from the extracellular solution by centrifugation through a 1:1.1 mixture of water-impermeable silicone oils barrier (AR 20 and AR 200) and frozen at −80 °C. For ofloxacin, cell sediments were suspended in 400 μl of 0.1 M glycine hydrochloride (pH 3) and lysed by incubating overnight at room temperature with light agitation. Cellular debris was removed by centrifugation (10 min at 14,000 rpm or 16,873*g*) and ofloxacin concentrations were analyzed by measuring fluorescence (ex. 292 nm, em. 496 nm) using a Synergy Mx Microplate Reader (BioTek). Ofloxacin concentrations were determined from a calibration curve for concentrations 0–300 ng ml$^{-1}$. For amikacin, an agar well diffusion bioassay was used[140]. Cell

sediments were suspended in 750 μl of PBS and lysed by incubation at 100 °C for 7 min. After centrifugation (10 min, 14,000 rpm or 16,873g) 100 μl supernatants was pipetted in 8 mm diameter holes on MHB agar plates inoculated with *Bacillus subtilis* ATCC 6051 as the indicator organism. Plates were incubated for 24 h at 37 °C after which the diameter of the inhibition zone was measured. Amikacin concentrations were deduced from a standard curve spanning a two-fold concentration range from 2 to 64 μg ml$^{-1}$.

**Ratiometric measurement of ATP:ADP ratios and cytoplasmic pH.** ATP:ADP ratio was determined using the ratiometric GFP-based biosensor Perceval[64]. The pRsetB-his7-Perceval plasmid was obtained from Addgene (plasmid #20336) and the perceval gene was extracted by PCR using primer SPI-10577 and SPI-10578 (Supplementary Data 4). Next, Perceval was subcloned in pBAD/*Myc*-HisA, linearized through PstI and EcoRI digestion, using Gibson assembly (NEB)[15]. The 490/405 nm excitation fluorescence ratio (em. 530 nm) correlates with the ATP:ADP ratio and was determined using a Synergy MX Microplate reader (BioTek). Perceval's sensitivity towards intracellular pH was countered by resuspending cells that were growing on MHB supplemented with 2 g L$^{-1}$ arabinose and 100 mg L$^{-1}$ ampicillin, in M63 medium buffered at pH 7 supplemented with potassium benzoate (40 mM) and methylamine hydrochloride (40 mM) before measurement.

Intracellular pH was measured with a GFP-based ratiometric sensor called pHluorin. We transformed the strains of interest with the pGFPR01 or pNTR-SD-pHluorin vector in which pHluorin is expressed from the arabinose-inducible or IPTG-inducible promoters, respectively, $P_{BAD}$ and $P_{tac}$[65,141]. To induce pHluorin expression, cells were either grown on MHB medium supplemented with 2 g L$^{-1}$ arabinose or 1mM IPTG or on M9 medium supplemented with 5 g L$^{-1}$ glucose and 20 g L$^{-1}$ arabinose with ampicillin (100 mg L$^{-1}$) to select for the maintenance of the plasmid. Note that no ampicillin and arabinose was added after nutrient shift. Ratiometric fluorescent measurements to determine pH were performed by either a Synergy MX Microplate reader (BioTek; ex. 410 and 470 nm ±20 nm; em. 530 nm ±20 nm) or a Spark plate reader (Tecan; ex. 380 and 470 nm ±20 nm; em. 530 nm ±20 nm). Cells were either resuspended in M63 minimal salts medium or spun through HPLC spin filters (BaseClear) and resuspended in fresh M9 medium before measurement. A calibration curve was generated by measuring fluorescence ratios of cultures resuspended for at least 5 min in a range of buffers with different pHs supplemented with 40 mM potassium benzoate and 40 mM methylamine hydrochloride to dissipate the transmembrane pH gradient (see Martinez et al., 2012). The relation between the fluorescence ratio and the intracellular pH was described by fitting a Boltzmann sigmoid:

$$ratio = Bottom + (Top - Bottom)/(1 + exp((V50 - pH_i)/Slope)).$$

**Image analysis.** Analysis of pictures of protein or DNA gels was done using Vilber's VisionCapt software for quantification and to estimate sizes.

**Dynamic proteomics.** For the dynamic proteomics analysis, we followed previously published protocols with minimum modification[26,142]. Briefly, cells were grown in M9 glucose medium with regular dilutions to maintain growth in exponential phase. At OD$_{600}$ 0.45, a proteomics sample was collected, and persisters were generated by a switch to M9 fumarate medium. At multiple time points after the nutrient switch, samples were collected, all containing $3 \times 10^8$ cells. They were centrifuged and washed with phosphate buffered saline (2×) and then cell pellets were frozen in liquid nitrogen. Cell sediments were lysed in 2% sodium deoxycholate, 0.1 M ammonium bicarbonate and disrupted by two sonication cycles (Hielscher ultrasonicator). BCA assay (Thermo Fisher Scientific) determined the protein concentration. Sample preparation involved: reduction with 5 mM TCEP (10 min, 95 °C), alkylation with 10 mM iodoacetamide (30 min, in the dark at room temperature), quenching with 12.5 mM N-acetylcysteine, dilution with 0.1 M ammonium bicarbonate to a concentration of 1% sodium deoxycholate, digestion with trypsin (Promega; overnight with 50:1 protein:trypsin ratio), supplemented with 0.5% TFA and 50 mM HCl, removal of precipitated sodium deoxycholate (15 min at 4 °C at 21,000g), desalting of peptides (C18 reversed phase spin columns; Macrospin, Harvard Apparatus), drying under vacuum, and storage at −80 °C until further processing. Samples were run on a dual pressure LTQ-Orbitrap Velos mass spectrometer connected to an electrospray ion source with peptide separation through an EASY nLC-1000 system (all Thermo Fisher Scientific) equipped with a RP-HPLC column (75 μm × 45 cm) packed with C18 resin (ReproSil-Pur C18–AQ; Dr. Maisch GmbH) using a linear gradient from 95% solvent A (0.15% formic acid, 2% acetonitrile) to 28% solvent B (98% acetonitrile, 0.15% formic acid) over 90 min at 0.2 μl min$^{-1}$. The acquisition mode obtained one high-resolution MS scan in the FT part at a resolution of 120,000 full width at half maximum (at m/z 400) followed by MS/MS scans in the linear ion trap of the 20 most intense ions. The charged state screening modus excluded unassigned and singly charged ions (dynamic exclusion duration: 20 s; ion accumulation time: 300 ms (MS) and 50 ms (MS/MS)). The raw files were imported into the Progenesis LC-MS software (Nonlinear Dynamics, Version 4.0) and analyzed using the default settings. MS/MS-data were exported in mgf format and searched against a decoy database of the forward and reverse sequences of the predicted proteome from *E. coli* (Uniprot, download date: 15/6/2012, total of 10,388 entries) using MASCOT. In the search criteria, full tryptic specificity was required (after lysine or arginine

residues), three missed cleavages were allowed, carbamidomethylation (C) was set as fixed modification, oxidation (M) as variable modification and mass tolerance was 10 ppm for precursor ions and 0.6 Da for fragment ions. Results were imported into Progenesis, and the false discovery rate (FDR) was set to 1%. The final protein lists containing the cumulative peak areas of all identified peptides for each protein, respectively, were exported from Progenesis LC-MS and further statically analyzed using an in-house developed R script (SafeQuant)[142]. The raw mass spectrometry data is available through the ProteomeXchange Consortium (ID: PXD029006).

For the analysis, we normalized our data relative to a 2-fold difference in protein concentrations between all the analyzed conditions with the WT at start (before nutrient shift) as a reference. Next, we applied unscaled, two-dimensional principal component analysis using the FactoMineR package (R). For this, we used a previously published and analyzed dataset (from Schmidt et al.[69] and Fig. 4a in Radzikowski et al.[26], ProteomeXchange IDs: PXD000498 and PXD001968, respectively) to generate the two dimensions that constitute the PCA space. We plotted our data on this space to compare with previously found trends.

**Radioactive assessment of translation activity.** To probe protein synthesis, we opted to follow the incorporation of radioactive L-(4,5-$^3$H)-leucine during stationary phase in complex MHB medium (±18 h after 1:100 dilution as done during the evolution experiment) instead of following induction of a fluorescent protein as done before[8] as the latter might be prone to biased results. Specifically, L-(4,5-$^3$H)-leucine was added to stationary phase cultures at 2.5 μCi ml$^{-1}$ and further incubated shaking at 37 °C. As control, chloramphenicol was additionally added at 64 μg ml$^{-1}$, which fully abrogated protein synthesis (data not shown). Over the course of 3h, samples were taken on an hourly basis and precipated in ice-cold trichloric acid (TCA; 10%). Precipitates were washed twice with ice-cold distilled water and added to scintillation liquid (Ultima-Flo M, Perking Elmer). Radioactive signal was measured as counts per minute with a Hidex 300SL scintillation counter.

**HPLC-UV detection and quantification of ppGpp.** Solutions containing ppGpp (Jena Biosciences) and products of its hydrolysis were analyzed using a HPLC-UV method measuring absorbance at 260 nm. A PL-SAX anion exchange column (PL-SAX 1000Å 8 μm, 50 × 4.6 mm, Agilent) was used for sample separation at 60 °C. Two methods were used: either (1) 0–2 min linear gradient from 80% to 70% A, 2–4 min linear gradient from 70% to 0% A, 4–9.5 min 0% A, at 9.5 min step change to 80% kept until 10 min; or (2) a 2.5 min (or 5 min, respectively) isocratic flow of B; A: 0.01 M K$_2$HPO$_4$, pH 2.6; B: 0.5 M K$_2$HPO$_4$, pH 3.5. The flow rate was 1 ml min$^{-1}$ in (1) or 1.2 ml min$^{-1}$ in (2).

**Spontaneous ppGpp hydrolysis assays.** Spontaneous (i.e. non-enzymatic) ppGpp hydrolysis can occur at low pH once the two distal phosphate groups of ppGpp are deprotonated and a positively charged divalent metal ion coordinates between the phosphate groups (Supplementary Fig. 6a). Spontaneous hydrolysis assays were performed in a buffer containing 0.05 M Tris, 0.09 M sodium formate and 0.025 M ammonium acetate, with adjusted to pH values and manganese concentrations as indicated. ppGpp (Jena Biosciences) was added in a final concentration of 1 mM sampled by HPLC at 37 °C. In order to prevent evaporation, the mixture was covered with PCR mineral oil. 2 μL of the reaction volume was injected at different points in time and the amount of ppGpp, as well as its hydrolysis products, were quantified by comparison to pure standards at different concentrations. Consistent with computational estimations of the pK$_a$ values of the two distal phosphate groups of ppGpp[143] (i.e. 6.5 and 6.8), we found that spontaneous ppGpp hydrolysis is high at pH of 8, and absent at pH of 7, with a sharp change in velocity around pH 7.5, and only occurs in the presence of divalent metal cations like manganese (Supplementary Fig. 6b-d, g). Thus, at pH values found in persisters, spontaneous ppGpp hydrolysis is inhibited (Fig. 5a, b).

**Overproduction and purification of 6His-SpoT.** To identify whether an enzyme-based catalysis of ppGpp hydrolysis (potentially also pH inhibited) is partially responsible for the previously observed increase in ppGpp levels in persisters[26], we focused on the enzymes synthesizing (RelA, SpoT) and hydrolyzing ppGpp (SpoT). As a deletion of *relA* had no effect on persister fractions, we had previously inferred that SpoT must be the player responsible for increased ppGpp levels in nutrient shift-induced persisters[26]. To further identify whether the ppGpp increase is due to an increased SpoT synthetase activity or a decreased SpoT hydrolysis activity, we initially analyzed a strain carrying a SpoT variant, which is defective in (p)ppGpp synthesis, but not in hydrolysis activity (SpoT-E319Q)[144]. After a glucose-to-fumarate nutrient shift, we found the same fraction of persisters as in the wild type (Supplementary Fig. 6e), suggesting that an inhibition of the hydrolase activity of SpoT must be responsible for the increased ppGpp levels in persisters formed upon a nutrient shift. To test whether, next to the spontaneous ppGpp hydrolysis, also the SpoT hydrolase activity is pH dependent, we aimed to purify SpoT, whose production and purification is notoriously difficult[145,146]. While the protein mostly resided in the insoluble protein fraction, we could purify minute amounts of soluble protein after a short induction time using either a Tris-based buffer (Supplementary Fig. 6h) or a cytosolic buffer (phosphate-based buffer, see below and Supplementary Fig. 6j).

SpoT was expressed with an N-terminal His-tag using plasmid pETM11-SpoT. pETM11-SpoT was generated by amplification of *spoT* using primer pair spoT_NcoI_s and spoT_NotI_stop_as (Supplementary Data 4) using genomic DNA from *E. coli* BW25113 as template. Subsequently the DNA fragment was cloned into pETM11 (EMBL Heidelberg) using NcoI and NotI restriction sites. Successful insertion was confirmed by restriction and sequencing analyses. The plasmid was transformed into *E. coli* BL21 (DE3) pLysS and selected on LB plates containing kanamycin and chloramphenicol.

*E. coli* strain BL21 (DE3) pLyS transformed with plasmid pETM11-SpoT was grown aerobically at 37 °C in 100 ml LB medium in a 1000 ml flask supplemented with kanamycin and chloramphenicol (both at 50 µg ml$^{-1}$). Gene expression was induced at $OD_{600} = 0.5$ by adding 0.5 mM isopropylthio-β-galactoside (IPTG) and cells were grown for 1 h at 37 °C. Cells were harvested, washed with buffer (50 mM Tris/HCl pH 7.5, 10% glycerol) and disrupted by passage through a Cell disruptor (Constant Cell Disruption Systems, Northants, UK) at 25 bar and 4 °C in disruption buffer (50 mM Tris/HCl pH 7.7, 10% (v/v) glycerol, 300 mM NaCl, 50 mM KCl, 1 mM dithiothreitol, 0.5 mM PMSF, and 0.03 mg/ml (w/v) DNase). After removal of intact cells and cell debris by centrifugation (3000g, 10 min, 4 °C), the cytosol was incubated over night at 4 °C with pre-equilibrated Ni-Sepharose (with 50 mM Tris/HCl pH 7.7, 10% (v/v) glycerol, 300 mM NaCl, 10 mM MgCl$_2$, 2 mM β-mercaptoethanol, 10 mM imidazole). Unbound protein was removed by washing with equilibration buffer and 6His-SpoT was eluted from the column by increasing imidazole concentrations up to 250 mM. Obtained purified protein degraded already at 4 °C (Supplementary Fig. 6i).

To obtain more stable protein, purification was attempted with cytosolic buffer. Cells were cultivated as described above and disrupted in cytosolic buffer pH 7.5 (6 mM KH$_2$PO$_4$, 14 mM K$_2$HPO$_4$, 140 mM KCl, 5.5% glucose, 10 mM NaCl) containing 0.03 mg/ml DNAse (w/v) and 0.5 mM PMSF. Removal of intact cells and purification of 6His-SpoT was performed as described above, but with use of cytosolic buffers containing imidazole. Note that pH of the buffers was re-adjusted after imidazole addition using phosphoric acid and KOH, respectively. For purification of SpoT at different pH, cells were disrupted in cytosolic buffer pH 7.5, bound to Ni-Sepharose and washed with buffer pH 7.5, and eluted with buffers adjusted to the corresponding pH and containing increasing imidazole concentrations. Protein content in samples was analyzed by SDS PAGE (12%) and Bradford protein assay[147]. The purified protein was slightly more stable than with the Tris-based buffer, but ppGpp hydrolysis activity declined strongly over time (Supplementary Fig. 6c, d). In addition, the amount of protein was rather low (0.3 mg ml$^{-1}$) and attempts to concentrate the samples or exchange buffers remained unsuccessful, rendering systematic assessment of the pH dependency of SpoT hydrolysis impossible in this way.

**Co-expression and purification of encapsulin and SpoT**. To obtain purified and stable protein, we followed an alternative approach, where we expressed SpoT in nanocages, formed as icosahedral capsids of 60 monomers of a small bacterial protein (encapsulin), which can protect the cargo protein from aggregation and degradation[148]. Nanocages or encapsulins are proteins that form macromolecular assemblies, resembling viral capsids[148]. Such capsids usually contain DyP-peroxidase or ferritin-like proteins[148,149]. Efficient encapsulation in vivo is possible due to the presence of ~30 AA C-terminal signal peptide, which interacts with inner surface of encapsulin[148]. Genes encoding for encapsulin and cargo enzymes are found together in an operon, since encapsulin assembly and cargo loading is a concerted process[148]. Here, we used a newly discovered robust encapsulin from *Mycolicibacterium hassiacum*, for which it was demonstrated to increase the stability of foreign cargo enzymes[150] and which enabled the characterization of the activity of the encapsulated and thus stabilized enzymes, as the nanocages contain three types of pores (5-9 Å), which allow small substrates to enter or leave the capsule[148,149,151].

For co-expression of SpoT and encapsulin, *spoT* was amplified using forward and reverse primer (Supplementary Data 4). BsaI sites are underlined and specific overhangs are indicated in bold. *spoT* was cloned in a pENC vector based on pET28 with a changed origin of replication to p15A in order to be compatible with pBAD vector, and with an introduced BsaI restriction sites. These changes have been introduced using Gibson assembly. Briefly, the p15A ori was amplified from pEVOL-pAzF (Addgene plasmid #31186) and the pET-28a(+) vector was amplified excluding pBR322 ori. These two fragments were assembled using NEB's Gibson assembly mix, with all the primers designed using Geneious software (Gibson cloning option; vector map available on request). The gene encoding SpoT was cloned into pENC in such way that, when expressed, it contains a N-terminal 6xHis-tag and also C-terminal fusion tag of 30 amino acids from DyP-peroxidase from the same operon as the original encapsulin gene[150]. EncMh encapsulin was provided by expression from pBAD-EncMh, obtained by cloning the codon-optimized gene encoding EncMh (GenScript) between the NdeI and HindIII restriction site of a pBAD-NK vector[150]. After sequencing, both the vectors were co-transformed into *E. coli* BL21-AI strain (ThermoFisher) and selected on LB plates containing ampicillin, kanamycin and tetracyclin.

*E. coli* strain BL21-AI transformed with plasmids pBAD-EncMh and pENC-spoT was grown aerobically at 37 °C in 200 ml LB containing ampicillin and kanamycin (both 50 µg ml$^{-1}$) in a 1000 ml baffled flask. At $OD_{600} = 0.5$ expression of *spoT* was induced by 1 mM IPTG (final concentration) and expression of

encapsulin by addition of 0.2% arabinose (final concentration) and cells were incubated for another 16–18 h at 30 °C. Cells were collected by centrifugation for 20 min at 6000 rpm (3099g) and 4 °C. Sonication was carried out in 50 mM TrisHCl pH 7.5 with 150 mM NaCl. As it can happen that cargo protein is expressed in higher amount than what can be encapsulated it is necessary to remove excess. Clarified cell-free extract was loaded on pre-equilibrated Ni-Sepharose column to remove excess of cargo protein which contains 6xHis-tag. Since encapsulin does not contain any 6xHis tag and due to its size, it comes in flow-through fraction, together with any cargo protein it may contain. Flow-through fraction was mixed on ice 1:0.75 with 10% PEG8000 in 100 mM Tris/HCl pH 7.5 with 2 M NaCl and left on a nutating shaker at 4 °C for 8 h. Precipitated proteins were collected by centrifugation for 30 min at 4000 rpm (1377g) and 4 °C. Sediment was resuspended in 20 ml of 50 mM TrisHCl pH 7.5 and left overnight on a nutating shaker at 4 °C to re-dissolve completely. Re-dissolved ENC-spoT was aliquoted, flash-frozen in liquid nitrogen and stored at −80 °C. Following this approach, we could obtain co-purified encapsulin with stable SpoT at 3 mg ml$^{-1}$ (Supplementary Fig. 6m).

**Enzymatic ppGpp hydrolysis assay**. We used both the encapsulated SpoT and regularly purified SpoT to characterize the pH dependency of its ppGpp hydrolysis activity. While the product of the spontaneous hydrolysis reaction leads to GTP, experiments with the limited amounts of the regularly purified SpoT indicated that SpoT hydrolyzes ppGpp to GDP, and that manganese ions are necessary for hydrolysis (Supplementary Fig. 6g), consistent with earlier reported requirements of Rel/Spo homologues for hydrolysis[152,153]. Enzymatic ppGpp hydrolysis assays with 6His-SpoT and encapsulin SpoT were performed in cytosolic buffer containing 250 mM imidazole (and 0.1% triton-X100 in case of encapsulin SpoT) adjusted to the corresponding pH with a final protein concentration of 0.1 and 0.5 mg ml$^{-1}$, respectively. Unless stated otherwise, manganese was added (1 mM) and the reaction was started by adding ppGpp (1 mM). Samples were incubated at 25 °C and, at different time points, aliquots were taken in which the reaction was stopped by adding EDTA (10 mM). Two microliters of each aliquot was used for HPLC analysis as described above. Here, similar to the results of the spontaneous hydrolysis, we found that the hydrolysis activity is strongly pH-dependent: hydrolysis starts at pH values higher than 7.0, sharply increases around pH 7.5 and reaches a maximum at around pH 8.3, above which the hydrolysis activity drops again likely due to enzyme denaturation (Supplementary Fig. 6f). A control experiment with an empty capsule sample at pH 8 revealed no ppGpp hydrolysis and few limited experiments with the minute amounts of regularly purified SpoT confirmed the pH dependency of hydrolysis (Supplementary Fig. 6f). Thus, also the ppGpp hydrolysis of SpoT is strongly pH-dependent, with hardly any activity around pH 7.0 and very much increased rates above pH 7.5. As persister cells have a cytoplasmic pH below 7.5 (Fig. 5a, b), we conjecture that the cytoplasmic pH, via inhibition of ppGpp hydrolysis, is a major regulator of ppGpp levels upon strong perturbation of metabolic homeostasis.

**Statistics**. Data were analyzed using either GraphPad Prism 8 or R in the RStudio environment. Relevant information regarding the statistical tests was added to each figure caption where appropriate. In Fig. 1b, c, Supplementary Figs. 1f–I and 2a, b, Chi² goodness-of-fit tests with 200,000 Monte Carlo simulations to compute p-values were used to compare the observed distribution of our mutations counts in various categories to the expected probabilities based on random occurring mutations. More detailed information can be found under the section explaining the genome-wide sequencing analyses and Chi² statistics are reported in Supplementary Data 4. For ANOVAs and linear (mixed) models, assumptions of homoscedasticity and normality of the residuals were checked visually comparing fitted value with residuals and comparing predicted with actual residuals. Where relevant, the Brown-Forsythe test was used to test whether SDs were equal, the Variance Inflation Factor was used to estimate the degree of multicollinearity and a Chi² test was performed to test the effectiveness of including random effects/matching. Post hoc tests (posttest) were performed to make and correct for multiple comparisons. All tests for which it applies were two-tailed. Throughout the manuscript, a significance level cutoff of alpha = 0.05 was used. For visual purposes, asterisks denote significance levels as defined in the figure captions and each exact p value can be found in Supplementary Data 4. In addition, test statistics and degrees of freedom are reported in Supplementary Data 4.

**Reporting summary**. Further information on research design is available in the Nature Research Reporting Summary linked to this article.

## Data availability

The NGS Sequencing data generated in this study have been deposited in the SRA database under accession codes PRJNA498891, PRJNA498708, PRJNA498717, PRJNA270307, and PRJNA768774. Mass spectrometry raw data files have been deposited to the ProteomeXchange Consortium via the PRIDE partner repository under dataset identifier PXD029006. The data underlying all the figures generated in this study are provided in the Source Data file. Mass spectrometry data used in this study are available via the ProteomeXchange Consortium via the PRIDE partner repository under the dataset identifiers PXD000498 and PXD001968. Sequence data used in this study are

available in the NCBI nucleotide database (NC_000913.3, NC_007946.1, NC_007941.1), Uniprot database (P0AFC3, P0AFC7, P33599, P0AFD1, P31979, P33602, P0AFD4, P0AFD6, P0AFE0, P0AFE4, P33607, P0AFE8, P0AFF0), PDB database (6g2j, 4he8) and PubChem database (135398637). Source data are provided with this paper.

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

## Acknowledgements

The authors thank Susan Gottesman for the AB037 strain, Joan L. Slonczewski for the pHluorin plasmid, Steven B. Vik for the BA14 strain and Henri de Greve for the UTI89 strain. We further thank Ying Liu for help with cloning the plasmid for encapsulin-SpoT expression, Franziska Number for providing controls for the ACMA quenching, Alexandros Papagiannakis for valuable suggestions as well as for help with microscopy and Bert Poolman for valuable discussions. B.V.D.B. is recipient of fellowships from the Fund for Scientific Research, Flanders (FWO; 12O1917N, 12O1922N, V428917N, and 11C6812/4N), from the Federation of European Microbiological Societies (FEMS; RG-2016-0052), from the Belgian American Educational Foundation (BAEF; 2016-E083) and from the European Molecular Biology Organization (EMBO; ALTF 344-2017). J.E.M. was granted a fellowship by the Agency for Innovation by Science and Technology (IWT). This work was furthermore supported by grants from FWO (1528318N, G0B2515N and G055517N), KU Leuven (C16/17/006), the Flanders Institute for Biotechnology (VIB) (to J.M. and B.V.D.B), the Deutsche Forschungsgemeinschaft (278002225/RTG 2202) (to T.F.) and from the Dutch Research Council (NWO; VIDI grant 864.11.001) to M.H.

## Author contributions

Conceptualization, B.V.D.B., H.S., J.L.R., M.F., T.F., J.M. and M.H.; Methodology, B.V.D.B., H.S., J.E.M., J.L.R., A. S., M.F., T.F., J.M. and M.H.; Investigation, B.V.D.B., H.S., J.E.M., T.E.P.K., J.L.R., T.M., N.L., J.S., S.V., L.D., A.S.; Writing—Original Draft, B.V.D.B., H.S., J.L.R., J.M. and M.H.; Writing—Review & Editing, B.V.D.B., H.S., J.M. and M.H.; Funding Acquisition, B.V.D.B., T.F., J.M. and M.H.; Resources, A.S.,T.F., J.M., and M.H.; Supervision, J.M. and M.H.

## Competing interests

The authors declare no competing interests.
