## [Peer Review File · Nature Communications]

Mutations in respiratory complex I promote antibiotic persistence through alterations in intracellular acidity and protein synthesisREVIEWER COMMENTS

Reviewer #1 (Remarks to the Author):

In this study, Van den Bergh et al. explore how persister cell populations form via perturbations of metabolic homeostasis. Specifically, the authors follow-up on their previous work in evolving increased antibiotic persistence by identifying the *nuo* operon as a key evolutionary target in this process. The authors then proceed to characterize why mutations in this operon result in increased persistence. To do this, the authors use structural insights about respiratory complex I, the complex encoded by *nuo*, to hypothesize that mutations conferring increased persistence are associated with the proton translocation properties of the complex. Follow-up on this hypothesis using a number of methods resulted in a model where mutations in respiratory complex I contribute to a decreased intracellular pH that results in increased persistence via induction of RpoS and halting protein synthesis.

The challenge of understanding precisely how perturbations of metabolic homeostasis result in antibiotic persistence is interesting and important. In working on this challenge, this study brings a valuable perspective by conducting a mechanistic follow-up on laboratory evolution experiments. In doing so, the authors identify how specific mutations may confer increased antibiotic persistence, as well as potentially more general insights into the formation of persister cells. A few questions and concerns remain in interpreting the results and claims presented in this manuscript. These are presented below:

Major comments:

1. Given the high variation of the cytoplasmic pH data in 5A, it is notable that there are far fewer replicates in S4I and S4J. This makes the claim in line 282 about “no significant overall changes” questionable, particularly since the ‘M’ mutant does have a significant change based on the current data. As a result, it would be good to see more data here.
2. While the data presented in 5C does indicate that a *nuo* mutant displays a stronger acidification than a wild-type population (as claimed in line 303), it is less clear that this data supports the claim that *nuo* mutants display a stronger acidification than wild-type ‘persisters’ as claimed in line 32, line 272, and line 306. This uncertainty comes from comparing the cytoplasmic pH values for nutrient-shifted cells in 5B vs 5C where the *nuo* mutant in 5C reaches (what looks to be) a similar pH value as the nutrient-switch population in 5B. Furthermore, if one compares the +*rpoS*,-CAM data for WT and *nuoN* in 6F, it appears that *nuoN* may have a substantially larger persistent fraction in what appears to be the same experimental conditions as in 5C. This makes drawing this specific conclusion about wild-type persisters based on the population-level data in 5C difficult. These challenges should be addressed in order for that claim to be fully supported.
3. The proposed model (figure 7) suggests that the drop in pH (facilitated by compromised *nuo*) independently results in increased RpoS response and halting of protein synthesis. However, since RpoS appears to be critical to the *nuo* mutant phenotype (6C) and nonN Δ *rpoS* cells appear to still have active translation/degradation (inferred from 6D), this raises the question of what role the RpoS response is playing in halting protein synthesis and degradation. While a full investigation of this question may be out of scope, it would be interesting to see, for instance, the experiment in 5C conducted on nonN Δ *rpoS* cells to see if the RpoS response plays a role in the further acidification of the *nuo* mutant cell population.

Minor comments:

1. Line 90-91: This last sentence could be clarified. In the current construction it is unclear what “it” refers to in “connect it with...”
2. Line 122: The data supporting the direct claim made here in the figure caption title is in supplementary figure 1 not in figure 1.
3. Line 233: “Evolved persistence mechanisms are independent...” implies that all evolved persistence mechanisms are “independent of impaired antibiotic uptake...”, however, this paper only investigates the persistence mechanisms behind evolved mutations in the *nuo* operon.
4. Line 234: The data presented here does not directly inform on what ATP levels look like, only the

ATP:ADP ratio.

5. Lines 328-331: The text here only refers to a single cell. Is this observation representative of a population/sub-population of cells with this phenotype?
6. Lines 403-405: This last sentence is somewhat confusing. While the data in 6C does seem to indicate that RpoS is not needed for tolerance when translation is halted, in the context of the nuo mutants it does appear that the RpoS response is needed in order to get the full protective effect of the nuo mutation. As a result, within the context of the nuo mutants (the topic of this sentence), the RpoS response appears to be critical (line 322) not 'unnecessary' (line 405).
7. Figure 1: "2 # of populations" text is confusing as to what it refers to.
8. Figure 4A, S4F: Even with the connecting lines on the figure, it is still difficult to clearly compare the different histograms.
9. Figure 2D, 2E, 6C, 6F, S2F, S2G, S2H, S2I, S4D, S4E: Make sure it is noted what antibiotic concentration was used in these experiments.
10. Figure 2D, 6C, 6F, S2I: Make sure it is noted what the treatment time was for these experiments.
11. Figure S4D: The legend should be corrected to indicate which drug is denoted by the checked shading.
12. Table S1: Many of the rows in this excel file were filtered. The default formatting should be to show all rows.
13. Table S3: Many of the rows in this excel file were filtered. The default formatting should be to show all rows.

Reviewer #2 (Remarks to the Author):

The work by Van den Berg et al. studies the mechanism for the enhanced tolerance of nuo mutants that appeared during experimental evolution under antibiotics. Based on their previous findings of enhanced survival at stationary phase due to mutations in the nuo genes, they now characterize in more details the reasons for the higher survival. First, focusing on three mutants, they show that the mutations results in impaired proton translocation activity. One of the potential reasons for higher survival to aminoglycosides that could be linked to impaired proton translocation is the uptake of the drug. The authors rule this out by measuring the uptake of a fluorescent dye and assays of the intracellular concentration of antibiotics. One common reason for antibiotic persistence is a decreased energy level. Indeed, the authors show that the ratio of ATP:ADP is clearly lower in the mutants. They show that this is not a sufficient condition for persistence by measuring similarly decreased energy levels in deletion mutants. They then measure the ability of the tolerant mutants to regulate their intracellular pH. They measure first the cytoplasmic pH in bulk populations and see a small decrease compared to the wt. Maybe the most significant result of the article is the strong link to RpoS for the nuo mediated tolerance. The authors also show that a protein translation decrease in the tolerant mutants. They go on showing that the decreased tolerance in the rpoS null strains (in wt or nuo background) is restored by CAM. It therefore seems that the main effect of the nuo mutations is the decrease in translation. The authors then perform a very nice analysis of the dynamics of persistence on previously defined PCA axes, showing the difference between the wt, the nuo mutants and the rpoS deletion mutants. This view confirms that the nuo mutations lead to a different adaptation to stationary phase.

The main comment is that the rpoS deletions observations, which are the most significant, are left for the latter part of the manuscript and not part of the title. In contrast, it seems that the link to pH decrease is weaker, as the pH decrease in the nuo tolerant mutants is less than 0.2. The authors build a model in which pH decrease is a central hub in persistence, triggering an rpoS mediated decrease in protein translation, but it is not clear whether the effect of pH is driving the phenomenon or just a by-product. One possible way to distinguish the pH effect would be to measure protein translation also in the nuo deletion mutants, where the pH does not decrease. The work is packed with very interesting observations but I feel it would be more convincing if centered more around the clear observation of rpoS role and presenting the pH effect only as an interesting possibility. Another main comment is that the way single persisters are defined, using a fluorescent marker but without a way to distinguish them from dead cells (which would have an affected pH) requires more careful

measurements.

Additional comments:

1. The authors write in the text generally that “nuo
2. The authors show that decrease ATP:ADP ratio occurs in the mutants but rule it out as a cause for persistence because it also occurs in conditions that do not lead to persistence, and go on looking for other reasons. However, it may be that this decreased energy level is a necessary condition, couldn't it? Have the authors shown that persistence can occur also without the decrease in ATP:ADP in their mutants?
3. In Fig. 5 A, the pH is measured at stationary phase which may change over time. Mutants that enter stationary phase late may have the same profile, but different values if sampled at the same time. It is therefore important to show the time dependence of the intracellular pH for the wt and the mutants over time, similarly to Fig. 5C.
4. Fig. 5B: it is surprising that growing bacteria in standard conditions reach a pH above 8, while the typical value is 7.2-7.8 even when external values are extremely different (see for example L Slonczewski et al. PNAS 1981) . Can the authors explain their value?
5. In Fig. 5B, the authors compared the pH of growing cells and stationary phase persisters. It is not clear what to authors mean by “growing cells”: are these cells taken from exponential phase or cells that grew after a lag (i.e in the same frames as the persisters ,as shown in Fig. 6A)?
6. In Fig. 5C, a nice time-dependence of the pH is shown. It seems that the survival to amikacin in these conditions is shown in Fig. 6F after 30 min. However, at this time there is no difference n pH yet between the mutant and the wt. Can the authors explain this?
7. How are persisters in microscopy defined? Is it clear that they are actually able to grow after they enter the dormant state? Can one distinguish them from dead cells? The rpos::mcherry marker is not enough to characterize persisters.
8. Fig. 6B: how many cells are quantified?
9. Fig. 4A: It is difficult to see properly the distributions. Also, for the deletion mutants, the persisters are a small fraction of the population so that in order to rule out that they do not have impaired uptake, it would be useful to plot the “survival function” (i.e. 1-cdf) of the distribution on a log scale.
10. What would be the prediction of the persistence level for the hipA7 mutant if rpoS is deleted?

Reviewer #3 (Remarks to the Author):

This study identifies and examines how a set of point mutations in the nuo electron. Transport machinery in E. coli lead to high persistence. The mutations, isolated during experimental evolution to a number of different (but related/?) antibiotics, affect the rate of proton translocation but are clearly not nulls for the Nuo activities. The authors find that these mutants, but not nulls, lead to lower cytoplasmic pH, and build most of their models around this observation, suggesting that low cytoplasmic pH may be a universal signal that can led to dormancy. While the lowered pH is shown here, the evidence that low pH causes the other phenotypes seen here is not fully convincing.

1. The first part of the manuscript, describing the evolution work and the resulting mutations, is difficult to follow. Clarification and simplification would help here, with more details to Materials and Methods. Among the questions:

- a. Why is UT189 a reference? What strain was actually used? Resource material says SX43, and references previous paper, but I could not find SX43 there. I gather this is a derivative of MG1655, since that is probably the genome reference, but that is never clearly stated. Method as for 2016 with UT189? This was unclear. It was complicated because Fig. S1B is labeled NC_000093.3, which doesn't exist. Should this be NC_000913.3?
- b. Table S1 and elsewhere: Were mutations in noncoding regions not noted at all, or are there really none that reached the 5% level?
- c. Table S3 shows that there are more mutations, at lower frequencies, in the selected genes. I gather from the text that all 29 of the target genes listed in J, for instance, were examined for extra mutations. The legend for Table S3 should be very clear about what genes were examined and also

whether there was any cutoff for frequency. It is not possible to tell from what was presented whether there are orders of magnitude more mutants everywhere, with some also in *nuo*.

d. Clarify what the carbon source was for growth in MHB media.

2. It would be very useful to understand to what extent the *nuo* alleles here are sufficient for the persistence phenotype (and other measured phenotypes). It should be possible, using the *yfbP::kan* linked marker (or another one), to transduce the mutant allele into the parental background. Does it should high persistence?

3. Lines 249-255, ATP:ADP ratios. This could be clearer. Maybe present the fact that ATP:ADP is down for both the points and the deletions, in both stationary (Fig. 4) and exponential (Fig. S4), and that this does not correlate with persistence. So low ATP:ADP is not sufficient to give high persistence; it doesn't really say that low ATP:ADP isn't necessary, does it? It would be useful to make that case clear, since it is, in my mind, qualitatively different from the data showing that antibiotic entry is unaffected. Here there is an effect, and it well may be important.

4. In Figure 6, the authors use a number of different protocols to lead to persistence, and it is frequently difficult to follow to what extent these are likely to have the same pathways and components. Does a *nuo* mutant further increase persistence in the *hipA* allele? After switch from glucose to fumarate? Or are each of these working in the same pathway? Are the authors suggesting, from data like that in Fig. 5C, that in a shift from glucose to fumarate, the fraction of cells entering a persister state depends on them decreasing pH further/for a longer time, or is the lower pH over time the effect of entering a persister state? What happens in an experiment like panel 5C, but with a *nuo* deletion that does not lead to high level persistence?

5. If RpoS levels are increased in some other manner, is that sufficient to lead to persistence, as implied by some of the work here?

6. Should Fig. S4J be labeled for internal pH?

7. The section of the paper on low pH and its relationship to persistence is not fully convincing.

Among the issues:

a. In Fig. 6B, one cell that became a persister was followed, if I read this correctly, and a correlation of lower pH and increased RpoS was seen. This is interesting, but hardly evidence for the model suggested here. This is not in a *nuo* mutant; how clear is it that the same process goes on in the *hipA* allele? In any case, one cell obviously is not proof of much. No other cells express either lower pH or higher RpoS (are these always paired?).

b. The implication in the text is that the effect of *nuo* is via the lower pH and thus induction of RpoS. Have the authors ruled out changes in signaling via the *ArcA/ArcB* two component system, regulated by electron transport and known to affect RpoS levels? What is the evidence that it is lower pH that triggers the RpoS-dependent downstream effects?

c. It seems important to examine the levels of RpoS, possibly with the *RpoS::mcherry* fusion, in the *nuo* cells.

d. In Fig. 6D, while it is clear that loss of *rpoS* has an effect in the *nuo* mutant on the proteome, what does the *rpoS* mutant alone do? It is a bit hard to evaluate the pattern here without that additional comparison. The Methods implies that after the fumarate switch, all cells are persisters. Is that the case? I could not find any of the actual proteins that are being monitored here. Is that all in previous publications? The primary data needs to be deposited somewhere.

8. *RelA* and *SpoT*: It is somewhat surprising that deletion of *relA* and *spoT* doesn't affect persistence, given that persistence is RpoS dependent, and usually RpoS levels are significantly decreased in the absence of all ppGpp. That led me to look at the construction of these double mutants, which generally are also quite sick.

a. The Resource list says P1 was made on CF1693, but the listing on the *nuo relA* spot strains says they were transduced with P1 (JW5437-1), a strain that is not listed, and cured of *kan*. Which is it? How were the strains confirmed?

b. Were RpoS levels measured at all in these double mutants?

c. If in fact *SpoT* and *RelA* are not necessary for the phenotype studied here, the section in the supplemental figures (Figure S5 and S6) on *SpoT* purification and effects of pH seems unnecessary. Why was *SpoT* included in Fig. 7, given the results here?

9. The authors suggest, based on the experiments in Fig. 6D, E, and F, that a decrease in protein synthesis is an outcome of the *nuo*-based persistence. Is the decrease in leucine incorporation seen in 6E reversed in an *rpoS* mutant? Is it not seen in a *nuo* deletion?

10. Figure 7: In this figure, it would seem that the lower protein synthesis is independent of RpoS induction. That doesn't seem to be what the authors say in the text. Is the suggestion that some RpoS but not too much is necessary for persistence? What is the evidence for decreased translation inhibiting RpoS induction, or is that an assumption if all new translation is halted?
11. What is the evidence for halted protein degradation? If some degradation is going on, those amino acids might support synthesis of some proteins, not detected by the leucine labeling used here.
12. Given the suggestion here that reduced translation is found in persistent cells, the authors may want to cite a recent paper finding that mutations in the translational apparatus give hyperpersistence (Khare and Tavazoie, 2020).

Reviewer #4 (Remarks to the Author):

The paper by Heinemann and colleagues is aimed at providing mechanistic understanding of how perturbation of metabolic homeostasis leads to formation of a persister state. In my view there is no generic mechanism/trigger as many factors can trigger persistence. The authors focus on the role of Complex I in triggering persistence in *Escherichia coli*. This is not a new finding and Roberto Kolter performed many studies in the early 1990s on GASP mutants (e.g. *JBact* 1993 175:5642-5647) and he uncovered complex I as being a pivotal player – not one of his papers is cited in this manuscript. I feel this is a major oversight by the authors and there needs to be some discussion about this. The authors propose a model centred around cytoplasmic acidification through mutations in complex I (reduced proton pumping) inducing the persister state through the activation of the RpoS regulon and shutdown of protein synthesis. More experimental details are needed to fully explain this model. Overall the paper is well written and presented and some interesting ideas are put forward by the authors. The experimental aspects of this manuscript are very impressive – a large number of high technical experiments have been performed. However, there are some important details missing that need clarification. I like the fact the authors are trying to come up with new mechanisms around persistence.

Comments to address:

1. Lines 99-103: it would be good for the readers to understand why the antibiotics were chosen for this study. The focus was largely on inhibitors of protein synthesis – why?
2. The passaging protocol needs more details: For example, lines 551-555 – is the concentration of amikacin (400 ug/ml) sub-MIC, MIC or x-fold MIC? The same question applies to the other antimicrobials used in the study. What is the number of CFU/ml in the stationary phase culture for MHB? (Figure 1A)
3. Expression of the *nuo* operon. It has been reported that the *nuo* operon is regulated in response to various electron acceptors and oxygen availability. What is the pattern of *nuo* expression in the experimental setup used in this study to evolve mutants? For example, is the *nuo* operon expressed under these growth conditions (Fig. 1A)? A number of *nuo* mutants were characterised (lines 155-157) – do these *nuo* mutants showed enhanced survival in the absence of antimicrobials under stationary phase conditions? Do these mutation occur naturally in the absence of antimicrobial stress – see work of Kolter above. I was struggling to understand Fig. S1A - % survival is plotted in an unusual way in this Figure. Compare this to Figure 2D and 2E.
4. A key question in these *nuo* mutants is the expression of *ndh2* and cytochrome *bd* – do they switch to non-proton translocating complexes to maintain the *pmf*? Is there a change in the end product profile of the *nuo* mutants – can you detect fermentative end products?
5. Complex I (Figure 3). These experiments are very convincing but lack some important controls. I was unsure why the authors have not expressed the rates or NADH oxidation and oxygen consumption per mg of protein? Are all the different complexes purified and reconstituted to the same

level? Fig. 3A – what is the rate of NADH oxidation in the detergent micelle before reconstitution for the various complexes? Can these rates be calculated and reported.

6. Figure 3B: I am assuming this is inside-out vesicles (IMV) and therefore the rate should be reported per mg protein. If it is proteoliposomes, I would like to see the IMV data.

7. Figure 3C: shows that the proteoliposomes are leaky as the quench is reversed with time. This is not observed in the IMVs (Fig. 3D) – a control needs to be included to dissipate the delta pH to prove indeed it has been generated – could add CCCP. The data does indeed show impaired proton pumping but how does that compare with rates of NADH oxidation – are these mutants simply uncoupled i.e. some proton leak in the complex?

8. Line 239/Figure 4: it would be good to have some quantitative data on the membrane potential measurements. Same comment applies to delta pH measurements. The authors use CCCP as a control in Figure 4 – what happens when nigericin is used i.e. can you convert any delta pH into the membrane potential? How do the authors explain the increased uptake of amikacin in the 294D mutant (Fig 4B) – is the membrane hyperpolarised in this mutant? What is the total pmf in this mutant?

9. pH homeostasis Figure 5: a key piece of information missing in Figure 5 is that of the external pH values under the different conditions. This is crucial information, especially in the cells challenged with fumarate. *E. coli* is able to tolerate huge changes in intracellular pH with no effect on cell physiology. It has a battery of mechanisms to combat extreme intracellular acidity. The extremely modest changes in intracellular pH observed here are very difficult to reconcile when any profound effects on cell physiology. For example the “acidic” pH values are still around intracellular pH 7.0. This internal pH would have little effect on protein synthesis. If these mutants were truly defective in pH homeostasis an experiment should be performed in which the mutants are challenged with acidic pH in the range pH 5-7 and the internal pH measured. This would reveal a true defect in pH homeostasis – see PMID:11283297. One would argue that exposing wild-type cells to acidic pH to lower internal pH should also trigger persistence. Of course this would also reveal hyperpolarization of the membrane potential as the cells would lack the ability to interconvert the membrane potential into a delta pH (compromised proton pumping).

10. Figure 5: I need a clearer justification of the rationale for including fumarate in these experiments – is it being used as a carbon source or electron acceptor, which would imply hypoxia. What are the differences in extracellular acidification between glucose and fumarate? Have the authors tried any experiments under hypoxia – a stationary phase cell would be fairly hypoxic in this experimental setup. The authors might want to consider the review of Voskuil in their discussion (PMID: 30262111). Perhaps the alkaline pH in the wild-type explains the lack of tolerance?

REVIEWER COMMENTS

Reviewer #1 (Remarks to the Author):

In this study, Van den Bergh et al. explore how persister cell populations form via perturbations of metabolic homeostasis. Specifically, the authors follow-up on their previous work in evolving increased antibiotic persistence by identifying the *nuo* operon as a key evolutionary target in this process. The authors then proceed to characterize why mutations in this operon result in increased persistence. To do this, the authors use structural insights about respiratory complex I, the complex encoded by *nuo*, to hypothesize that mutations conferring increased persistence are associated with the proton translocation properties of the complex. Follow-up on this hypothesis using a number of methods resulted in a model where mutations in respiratory complex I contribute to a decreased intracellular pH that results in increased persistence via induction of RpoS and halting protein synthesis.

The challenge of understanding precisely how perturbations of metabolic homeostasis result in antibiotic persistence is interesting and important. In working on this challenge, this study brings a valuable perspective by conducting a mechanistic follow-up on laboratory evolution experiments. In doing so, the authors identify how specific mutations may confer increased antibiotic persistence, as well as potentially more general insights into the formation of persister cells. A few questions and concerns remain in interpreting the results and claims presented in this manuscript. These are presented below:

We would like to thank the reviewer for the spot-on summary about our work and the kind words, as well as the suggestions to improve it further. Since we made major changes covering large parts of the text, it would not result in a clear overview if we would fully cite all specific changes that we made for each of the comments listed below. Instead, we will indicate what we changed in response to each comment and refer to the line numbers of the revised manuscript without track changes where all the precise changes can be found when possible.

Major comments:

1. Given the high variation of the cytoplasmic pH data in 5A, it is notable that there are far fewer replicates in S4I and S4J. This makes the claim in line 282 about “no significant overall changes” questionable, particularly since the ‘M’ mutant does have a significant change based on the current data. As a result, it would be good to see more data here.

We agree that the previous data, especially on complex medium, carried significant noise. For this revision, as per suggestion of reviewer 1 and other reviewers, we have expanded our dataset on intracellular pH measurements and re-measured the pH_i while simultaneously assessing antibiotic tolerance levels. The higher number of independent repeats together with a more rigorous experimental procedures (e.g.: consistent, short time between sampling and measurement) resulted in a much clearer picture. Thereby, we could confirm that the M mutant shows no statistically significant change in intracellular pH in exponential phase and that in fact overall, no change is observed between the *nuo** mutants and the wild type under these conditions. The large changes in pH_i in stationary phase were re-confirmed and these new data can be found in Figs 5A, S4I and S4J.

2. While the data presented in 5C does indicate that a *nuo* mutant displays a stronger acidification than a wild-type population (as claimed in line 303), it is less clear that this data supports the claim that *nuo* mutants display a stronger acidification than wild-type 'persisters' as claimed in line 32, line 272, and line 306. This uncertainty comes from comparing the cytoplasmic pH values for nutrient-shifted cells in 5B vs 5C where the *nuo* mutant in 5C reaches (what looks to be) a similar pH value as the nutrient-switch population in 5B. Furthermore, if one compares the +*rpoS*, -CAM data for WT and *nuoN* in 6F, it appears that *nuoN* may have a substantially larger persistent fraction in what appears to be the same experimental conditions as in 5C. This makes drawing this specific conclusion about wild-type persisters based on the population-level data in 5C difficult. These challenges should be addressed in order for that claim to be fully supported.

As mentioned in our answer to the previous comment, we re-measured all pH_i data for all conditions, with more replicates and improved methods. This also includes the data that was shown in Fig. 5C. Here, we could confirm the stronger acidification in the *nuo** mutants compared to the wild type. As for the comparison of the data between the original Fig. 5B and 5C: please note that in Fig. 5B we showed pH_i data generated by microscopy and in Fig. 5C, we showed data obtained using a plate-reader. Comparing absolute pH_i values between microscopy and population-wide setups is not straightforward. In addition, the microscopy data was acquired 18-22 hours after the glucose-fumarate shift whereas population-wide measurements were performed dynamically immediately after the shift. In the latter condition, the wild type and *nuo** mutants show large differences in persister fraction, while we did not measure persister levels 22h after the shift.

In conclusion, to not confuse the reader, we decided to remove Fig. 5B (and all the microscopy data) as these data did not add much to our story and as other remarks on the microscopy data were raised by other reviewers. Overall, the improved measurements can be found in Figs 5B, S4L, S5A, S5D.

3. The proposed model (figure 7) suggests that the drop in pH (facilitated by compromised *nuo*) independently results in increased RpoS response and halting of protein synthesis. However, since RpoS appears to be critical to the *nuo* mutant phenotype (6C) and nonN $\Delta rpoS$ cells appear to still have active translation/degradation (inferred from 6D), this raises the question of what role the RpoS response is playing in halting protein synthesis and degradation. While a full investigation of this question may be out of scope, it would be interesting to see, for instance, the experiment in 5C conducted on nonN $\Delta rpoS$ cells to see if the RpoS response plays a role in the further acidification of the *nuo* mutant cell population.

We totally agree with the question raised by the reviewer, and also with the point that a full investigation would be out of scope. Yet, we performed the suggested experiments and additionally now also included tolerance measurements, proteome analyses and pH_i measurements on all strains with/without *rpoS*. In these additional experiments, we found that *nuo** strains lacking RpoS still show stronger acidification than the wild type (Fig S5D) and, in agreement with this acidification, that a deletion of *rpoS* only results in a minor decrease in tolerance (Fig 6A-B). Therefore, we prefer to classify the contribution of RpoS as a modest one. The proteome data also match these results. While the wild-type proteome, in the absence of RpoS, starts to adapt to fumarate upon a glucose-to-fumarate shift (see Fig 6C), the proteome of *nuo** mutants that acidify stronger seems to be frozen, regardless of the presence of RpoS (see Figs 6C-D, S5F, and S5G).

We are grateful that the reviewer suggested this experiment, which allowed us to further refine our model. Under slow and/or less dramatic acidification, internal acidification and RpoS support the activation of the stress regulon that leads to a higher tolerance towards antibiotics. Under fast/severe acidification, acidification alone is sufficient to halt protein turnover which leads to a sharp increase of antibiotic tolerance. We explain this now in the description of our model in the caption of Fig 7 and lines 446-452 and others.

Minor comments:

1. Line 90-91: This last sentence could be clarified. In the current construction it is unclear what “it” refers to in “connect it with...”

We have changed the wording of the sentence in our revised manuscript and can be found on lines 80-82.

2. Line 122: The data supporting the direct claim made here in the figure caption title is in supplementary figure 1 not in figure 1.

We have changed the claim in the title to be more descriptive and refer to the supplemental figure for the direct claim in the caption of Figure 1.

3. Line 233: “Evolved persistence mechanisms are independent...” implies that all evolved persistence mechanisms are “independent of impaired antibiotic uptake...”, however, this paper only investigates the persistence mechanisms behind evolved mutations in the *nuo* operon.

We agree and have rephrased the section title to: “Neither impaired antibiotic uptake nor decreased energy levels fully explains complex I-dependent increased persistence”

4. Line 234: The data presented here does not directly inform on what ATP levels look like, only the ATP:ADP ratio.

The reviewer is right. The used fluorescent sensor reports on the ATP:ADP ratio. Since ATP:ADP ratios better reflect the energy status (e.g.: Berg, 2009, Nature; Jiang, 2007, Biochemistry; Forchhammer, 2015, FEBS; Vander Heiden, 2009, Science; Bressan, 2020, eLife), it is important to note that our conclusion on the energy level of the *nuo** mutants remains valid. However, we have checked our manuscript and rephrased statements regarding ATP throughout the file to address more carefully that we measure ATP:ADP ratios as a representation of energy levels.

5. Lines 328-331: The text here only refers to a single cell. Is this observation representative of a population/sub-population of cells with this phenotype?

The reviewer is right. Observing this cell was a “lucky shot” which we made by accident: Essentially, we observed this single cell under the microscope dynamically entering the persister phenotype: The cell stopped growing, its cytosolic pH dropped, and it induced the *rpoS* promoter. Given the fact that persister cells are very rare and it would be difficult to find a sufficiently high number of such cells and given that the microscopy setup did not add much to our story, we opted to remove it from the revised manuscript to reduce the complexity of the story.

In the revised manuscript, we have included new additional evidence connecting persistence and cytoplasmic pH: First, we re-measured cytoplasmic pH through an improved assay at higher replication so that we can now report that the difference are indeed consistent and significant; Second, we now measured the pH_i and the antibiotic tolerance in the same experiment, where we find a strong negative correlation between cytoplasmic pH and antibiotic tolerance levels; Third and most importantly, in a new experiment where we artificially acidified the cytoplasm, we could establish a causal link between internal acidification and an increased persister level, thereby strengthening our initial claims. These data are contained within Figs. 5, S4 and S5 and the associated text and captions.

6. Lines 403-405: This last sentence is somewhat confusing. While the data in 6C does seem to indicate that RpoS is not needed for tolerance when translation is halted, in the context of the *nuo* mutants it does appear that the RpoS response is needed in order to get the full protective effect of the *nuo* mutation. As a result, within the context of the *nuo* mutants (the topic of this sentence), the RpoS response appears to be critical (line 322) not 'unnecessary' (line 405).

Our new data confirms a significant correlation between tolerance and pH_i in strains lacking *rpoS* and a modest decrease in tolerance in *nuo** mutants lacking *rpoS*. Therefore, we have now further specified that, although the presence of RpoS is contributing to increased persistence in the *nuo** point mutants, a full-blown stress response activated by RpoS - as is the case in the wild type (see Fig 6B and Radzikowski, 2016, MSB) - is not needed. These new data can be found in Figs 6A-C, S4K-L, S5A and S5D-E. Additionally we have rephrased our statements on this topic, mostly on lines 365-371 and elsewhere in the manuscript.

7. Figure 1: "2 # of populations" text is confusing as to what it refers to.

The numbers in the bars of the circos plots refer to the number of independent populations in which the specific mutations were found. We now have changed this point in the legenda and explained this more carefully in the figure caption.

8. Figure 4A, S4F: Even with the connecting lines on the figure, it is still difficult to clearly compare the different histograms.

We have changed the representation of the cytometer data from regular histograms to plotting the "1-cumulative distribution function" which allows a better comparison of the different graphs, as also suggested by another reviewer. This representation now shows the percentage of cells with an DiBAC4(3) intensity above the fluorescence value specified in the x axis (Figs 4A and S4E).

The slight increased depolarization in the *nuo** mutants compared to the wild type (i.e. curves are a bit higher from a fluorescence level >10) - agreeing with their loss of ΔpH - is rather limited, likely as ΔpH contributes only slightly to the electrical gradient (Kaila, 2021, NRM; Krulwich, 2011, NRM). Moreover, numbers clearly do not concur with the differences we observe in antibiotic survival (see text on lines

291-293 for numerical explanation). We have further specified this in the text as well on lines 254-261 and in the figure caption of Fig 4A.

9. Figure 2D, 2E, 6C, 6F, S2F, S2G, S2H, S2I, S4D, S4E: Make sure it is noted what antibiotic concentration was used in these experiments.

We clarified the concentration of the used antibiotics and treatment time in the caption of all the relevant figures and panels.

10. Figure 2D, 6C, 6F, S2I: Make sure it is noted what the treatment time was for these experiments.

We now specify the duration of treatment in the caption of the relevant figures and panels.

11. Figure S4D: The legend should be corrected to indicate which drug is denoted by the checked shading.

We thank the reviewer for pointing out that the checked pattern was lost in finalizing the figures. We have now corrected the legend.

12. Table S1: Many of the rows in this excel file were filtered. The default formatting should be to show all rows.

We apologize for the inconvenience and have removed the filter in the supplemental files of the resubmitted manuscript so that all mutations are visible. Upon acceptance, the mutations will also be deposited in the CAMEL database (<https://camelatabase.com/>) where they can easily be browsed and compared to other mutations across evolution experiments.

13. Table S3: Many of the rows in this excel file were filtered. The default formatting should be to show all rows.

Again, we apologize for this error, see previous comment.

Reviewer #2 (Remarks to the Author):

The work by Van den Berg et al. studies the mechanism for the enhanced tolerance of *nuo* mutants that appeared during experimental evolution under antibiotics. Based on their previous findings of enhanced survival at stationary phase due to mutations in the *nuo* genes, they now characterize in more details the reasons for the higher survival. First, focusing on three mutants, they show that the mutations

results in impaired proton translocation activity. One of the potential reasons for higher survival to aminoglycosides that could be linked to impaired proton translocation is the uptake of the drug. The authors rule this out by measuring the uptake of a fluorescent dye and assays of the intracellular concentration of antibiotics. One common reason for antibiotic persistence is a decreased energy level. Indeed, the authors show that the ratio of ATP:ADP is clearly lower in the mutants. They show that this is not a sufficient condition for persistence by measuring similarly decreased energy levels in deletion mutants. They then measure the ability of the tolerant mutants to regulate their intracellular pH. They measure first the cytoplasmic pH in bulk populations and see a small decrease compared to the wt. Maybe the most significant result of the article is the strong link to RpoS for the nuo mediated tolerance. The authors also show that a protein translation decrease in the tolerant mutants. They go on showing that the decreased tolerance in the rpoS null strains (in wt or nuo background) is restored by CAM. It therefore seems that the main effect of the nuo mutations is the decrease in translation. The authors then perform a very nice analysis of the dynamics of persistence on previously defined PCA axes, showing the difference between the wt, the nuo mutants and the rpoS deletion mutants. This view confirms that the nuo mutations lead to a different adaptation to stationary phase.

We thank the reviewer for highlighting the relevance and impact of our results and for thorough and constructive evaluation of our manuscript. Since we made major changes covering large parts of the text, it would not result in a clear overview if we would fully cite all specific changes that we made for each of the comments listed below. Instead, we will indicate what we changed in response to each comment and refer to the line numbers of the revised manuscript without track changes where all the precise changes can be found when possible.

The main comment is that the rpoS deletions observations, which are the most significant, are left for the latter part of the manuscript and not part of the title. In contrast, it seems that the link to pH decrease is weaker, as the pH decrease in the nuo tolerant mutants is less than 0.2. The authors build a model in which pH decrease is a central hub in persistence, triggering an rpoS mediated decrease in protein translation, but it is not clear whether the effect of pH is driving the phenomenon or just a by-product. One possible way to distinguish the pH effect would be to measure protein translation also in the nuo deletion mutants, where the pH does not decrease. The work is packed with very interesting observations but I feel it would be more convincing if centered more around the clear observation of rpoS role and presenting the pH effect only as an interesting possibility.

We fully agree with the reviewer. The manuscript indeed contains diverse and interesting data sets, which are connected in a system-like manner where even dynamics play a role. In this revised version, we have managed to present the data in a way that readers can even better appreciate it. We feel that the presented system level-picture will get us closer to understand the phenotype of persisters.

As for the small pH_i changes: For this revised manuscript, we have further optimized our cytoplasmic pH measurements and have redone almost all pH measurements (see also below). On the basis of this, we confirmed the reported pH differences and we found them to be significant. While a difference of 0.3-0.6 pH units seems small, we would like to point out that pH is on a logarithmic scale, so difference of "only" 0.5 pH units already represents a 3.2-fold change in H⁺ activity. Furthermore, small changes in pH can have profound and broad effects on many proteins and metabolites at the same time (e.g.: in

stability, charge, folding state, mobility, redox equilibria, activity, interaction; e.g.: Orij, 2011, BBA; Kozlowski, NAR, 2017), which is the main reason why pH_i homeostasis is usually a tightly controlled process that is affected by many systems. Additionally, throughout various systems in life, modest changes in pH_i (± 0.5) were shown to have an important role in various aspects of physiology, e.g., in: bacterial chemotaxis (Demir, 2012, Biophys. J.); glucose-sensing in yeast (Isom, 2018, JBC) linking membrane biogenesis to metabolism (Young, 2010, Science) and controlling growth rate (Orij, 2012, Genome Biol.); O_2 saturation of hemoglobine in blood cells (the Bohr/Haldane effect Bohr, 1904, Skand. Arch. Physiol.; Christiansen, 1914, J. Physiol.); cancer and stem cell behavior (e.g., Persi, 2018, Nat. Comm.). With our current revised manuscript, we strengthen our claims that pH_i also has a big role in bacteria, more specifically in the important phenotype of antibiotic-tolerant persister cells.

In fact, we have performed new experiments where we not only find strong negative correlations between cytoplasmic pH (dynamics) and antibiotic tolerance levels (in two different setups across two different labs; see Figs 5C, S4K-L and S5A), but we also identified a causal link between internal acidification and the increased persister level (see Figs 5D and S5B), thereby strengthening our initial claims. Lastly, we have expanded our experiments with dynamic pH_i measurements on all strains with/without *rpoS* while simultaneously measuring the proteome. These experiments confirmed that *nuo** mutants with strong acidification, seem to display a frozen proteome whereas modest acidification in the wild type triggers a full-blown stress response.

As for mutants lacking *nuo*, we opted to omit them from the physiological measurements to reduce the complexity of the story. Furthermore, these strains show a very strong shifted metabolism making them inappropriate as comparison in these assays (see Fuhrer, 2017, MSB; Prüß, 1994, JBac). This became very obvious when we examined the global redox state in these strains, as seen in the figure below:

Here, resazurin reduction is followed in time (forming an increase in fluorescence). Mutants lacking *nuo* strongly differed in redox state from the wild type and *nuo** mutants that were highly similar, thus indicating a highly different metabolic state in mutants lacking *nuo* which would complicate using them as a comparison in physiological assays.

Given our new experimental insights (e.g., still significant correlation between tolerance and pH_i in strains lacking *rpoS* and a modest decrease in tolerance in *nuo** mutants lacking *rpoS*), we have now

further specified that, although the presence of RpoS is contributing to increased persistence in the *nuo** point mutants, a full-blown stress response activated by RpoS - as is the case in the wild type (see Fig 6B and Radzikowski, 2016, MSB) - is not needed.

With this new data, now also demonstrating the causality between acidification and persistence, we feel that we found an optimal positioning of our work. Yet to address the comment of this reviewer, we now better place the still important role of RpoS through the new data (in Figs 6A-C, 6E, S4K,L S5A and S5D) and by highlighting this also in the text (lines 360-371) and discussion (lines 446-449).

Another main comment is that the way single persisters are defined, using a fluorescent marker but without a way to distinguish them from dead cells (which would have an affected pH) requires more careful measurements.

We understand the point of the reviewer. In fact, the microscopy data were essentially redundant to the population-level data. Also, slightly different pH values (between the population-level measurements and the microscopy data; due to technical aspects) led to confusion with another reviewer. Thus, as these data did not add much and the manuscript is already complex and full of data, we removed these data in the revised manuscript.

Additional comments:

1. The authors write in the text generally that “*nuo*

The comment seems be partially lost. In the text, we have consistently referred to the high-persistence conferring point mutants in *nuo* as “*nuo**” mutants/variants.

2. The authors show that decrease ATP:ADP ratio occurs in the mutants but rule it out as a cause for persistence because it also occurs in conditions that do not lead to persistence, and go on looking for other reasons. However, it may be that this decreased energy level is a necessary condition, couldn't it? Have the authors shown that persistence can occur also without the decrease in ATP:ADP in their mutants?

We agree with the reviewer. While indeed some recent papers suggest that persisters have low ATP (Conlon, 2016, Nat Microbiol; Pu, 2019, Mol Cell; Shan, 2017, mBio; Wilmaerts, 2018, 2019, mBio, Mol Cell), in our earlier work, we have shown that persistence can indeed occur also without the decrease in ATP:ADP (Radzikowski, 2016, MSB) when cells are shifted to fumarate as carbon source.

As we did not formally exclude the possibility that low energy contributes to the high persistence of the *nuo** mutants, we have adjusted the text to make sure that we do not make that claim. We now state that low energy cannot fully or single-handedly explain the high tolerance but can still contribute to persistence on lines 277-281.

3. In Fig. 5 A, the pH_i is measured at stationary phase which may change over time. Mutants that enter stationary phase late may have the same profile, but different values if sampled at the same time. It is therefore important to show the time dependence of the intracellular pH for the wt and the mutants over time, similarly to Fig. 5C.

We fully acknowledge that measurements on single time-points could hide relevant information. However, the timepoint that was chosen agrees with the point in time that was used to assess antibiotic survival and that was used during evolution experiments which selected for the *nuo** mutants. While following pH_i dynamically in time upon entry in stationary phase could be interesting to further resolve dynamic properties of the lower pH_i in the *nuo** mutants, such an experiment is extremely demanding, highly impractical, and likely also prone to variations that are irrelevant to our current story (e.g., batches of complex medium that differ slightly). Therefore, instead, we opted to resolve the high variation in Fig 5A by repeating the measurements of pH_i in the complex medium and coupling these measurements directly in the same experiment to tolerance assays to correlate one with the other (Figs 5A, S4I-K and S5E). Here, we found that there is a strong negative correlation between cytoplasmic pH (dynamics) and antibiotic tolerance levels. Additionally, we also identified a causal link between internal acidification and the increased persister level (see Figs 5D and S5B), thereby strengthening our initial claims.

In order to further address the comment of the reviewer on the relevance of pH_i dynamics, we have performed additional experiments following pH_i dynamically in time pH under the more controlled nutrient shift from glucose to fumarate, which directly coupled to tolerance assays and proteomic analyses. These data are now shown in Figs 5B-C, 6C, S4L, S5A, S5D and S5F-G and clearly indicate that *nuo** mutants acidify stronger upon metabolic perturbations, which correlates with a stronger induction of tolerance and a 'frozen' proteome.

4. Fig. 5B: it is surprising that growing bacteria in standard conditions reach a pH above 8, while the typical value is 7.2-7.8 even when external values are extremely different (see for example L Slonczewski et al. PNAS 1981). Can the authors explain their value?

We agree that pH_i measurements tend to vary slightly between setups (e.g., microscopy and population-level) and certainly between labs, constructs, and strains. While our microscopy values were on the high-end side of what is found in literature, others have also reported average pH_i values around or above 8 for growing *E. coli* (e.g. Zilberstein et al, 1984; Martinez et al, 2012 or Zarkan et al, 2019). To reduce variations across our experiments (population-level experiments, microscope experiments), we opted to leave out microscopy data (see response to one of the previous comments). One other source of variation in our previous manuscript was the timing between sampling and measurement. In the revised manuscript, we remeasured all pH_i data in the most careful manner with a most accurate determination of the calibration curves. Here, we discovered that the experiment shown in Fig 5C suffered from using an incorrect calibration curve which also caused the pH values to be higher. All figures are now updated and the pH_i values fall in line with those found in literature.

5. In Fig. 5B, the authors compared the pH of growing cells and stationary phase persisters. It is not clear what to authors mean by “growing cells”: are these cells taken from exponential phase or cells that grew after a lag (i.e in the same frames as the persisters ,as shown in Fig. 6A)?

Growing cells in Fig 5B refers to cells exponentially growing on M9 glucose. These cells come both from cultures before shifting them to fumarate and from stationary phase cultures that were dilution in fresh medium to restart growth. As stated before, we opted to leave out these data from the revised manuscript.

6. In Fig. 5C, a nice time-dependence of the pH is shown. It seems that the survival to amikacin in these conditions is shown in Fig. 6F after 30 min. However, at this time there is no difference in pH yet between the mutant and the wt. Can the authors explain this?

We totally follow the reasoning of the reviewer. In the revised manuscript, we have expanded all the dynamic pH measurements and directly coupled them to survival assays or proteomics. The correlations between survival and pH_i (Fig S5A) were already significant from 20-40 min after the switch (the time when the treatment started) and became stronger afterwards. Additionally, the proteome of these highly tolerant strains also seems to stop shifting after this short initial period after the shift (Fig 6C). These new data are now included in Figs 5B, S5A and 6C and we adapted the text on lines 1450-1453 and elsewhere to accommodate these new insights.

7. How are persisters in microscopy defined? Is it clear that they are actually able to grow after they enter the dormant state? Can one distinguish them from dead cells? The *rpos::mcherry* marker is not enough to characterize persisters.

In the revised manuscript, we have decided to leave out these microscopy data (see one of the previous comments).

8. Fig. 6B: how many cells are quantified?

We were only able to detect one event that captured a cell switching in an actively growing culture. Observing this cell was a “lucky shot” which we made by accident: Essentially, we observed a cell dynamically entering the persister phenotype: The cell stopped growing, its cytosolic pH dropped, and it induced the *rpoS* promoter. Given the fact that persister cells are very rare, this was really a lucky observation. Without excessive experimentation, it is impossible to find more such cells. Yet, as this piece of data is just from a single cell, it would be very difficult to find more such cells and the microscopy did not add much to our story, we opted to remove it from this manuscript. In addition, we have now also included additional evidence for a causal connection between persistence and pH_i (Figs 5, S4 and S5).

9. Fig. 4A: It is difficult to see properly the distributions. Also, for the deletion mutants, the persisters are a small fraction of the population so that in order to rule out that they do not have impaired uptake, it would be useful to plot the “survival function” (i.e. 1-cdf) of the distribution on a log scale.

We would like to thank the reviewer for this helpful suggestion on an alternative visualization of our distributions. We have now represented all the cytometry data in such manner (see Figs 4A and S4E and below) and have updated the text accordingly to fully explain this way of representing the data.

The *nuo** mutants show a decreased electrical potential compared to the wild type (i.e. they show higher curves than the wild type in Figure 4A, see also below) - a logical consequence of the loss of proton translocation in complex I. However, the decrease in electrical gradient is minor - as expected, since the contribution of pH to the electrical gradient is usually only minor (Kaila, 2021, NRM; Krulwich, 2011, NRM). Furthermore, the effect is limited to a much smaller fraction of the population compared to the differences we observe in antibiotic survival. For example, the wild-type persister level of $\pm 0.1\%$ agrees with a DiBAC₄(3) fluorescence level of 650 and above (cyan lines). However, only <1% of the *nuo** mutants have such a high level of DiBAC₄(3), whereas the strains contain on average 50% surviving cells. Similarly, CCCP treatment depolarizes cells resulting in a fluorescence level of 30 and above (fuchsia lines). While such a treatment induces 100% survival (Kwan, 2013, AAC), a similar fluorescence level/depolarization status is only present in <3% of cells in the *nuo** mutants and 0.3% of cells in the wild type. A similar observation is also true for the *nuo** mutations in the uropathogenic background (Fig S4E).

Given our bioassay (no decreased aminoglycoside uptake) and the unchanged uptake of ofloxacin (Figure S4A) - a fluoroquinolone whose uptake is independent of electrical gradient (Aldred et al., 2014; Cramariuc et al., 2012; Vesselinova et al., 2016) and to which mutations in *nuo* also offer increased protection (Figure S2G) - our conclusion remains valid that a difference in antibiotic uptake does not explain the increased persister levels in the *nuo** mutants.

Thus, plotting the data according to the suggestion of the reviewer, shows a minor effect on the electrical gradient in the *nuo** mutants which cannot fully explain the increased tolerance in these

strains. In addition to the updated figures, we have updated the figure captions and relevant sections of our results to rightfully capture this information (e.g. lines 252-261).

10. What would be the prediction of the persistence level for the hipA7 mutant if rpoS is deleted?

Although such an experiment would certainly be interesting, we have not further analysed the hipA7 mutant for the following reasons: first, we want to keep the focus on *nuo* and pH_i , secondly, we have omitted the microscopy data (which was the only point with hipA7 data) and thirdly, the impact of RpoS in the revised story is modest.

Reviewer #3 (Remarks to the Author):

This study identifies and examines how a set of point mutations in the *nuo* electron transport machinery in *E. coli* lead to high persistence. The mutations, isolated during experimental evolution to a number of different (but related/?) antibiotics, affect the rate of proton translocation but are clearly not nulls for the Nuo activities. The authors find that these mutants, but not nulls, lead to lower cytoplasmic pH, and build most of their models around this observation, suggesting that low cytoplasmic pH may be a universal signal that can lead to dormancy. While the lowered pH is shown here, the evidence that low pH causes the other phenotypes seen here is not fully convincing.

We would like to thank the reviewer for the thorough evaluation of our manuscript and the useful feedback that we received. As shown below, we have substantially expanded our experimental data and analyses and have made major textual changes which further strengthen our claims. Most importantly, and addressing the comment of the reviewer, we added experiments where we measure pH_i and antibiotic tolerance in the same cells, showing a strong negative correlation between these two measures, and furthermore, we added an experiment which demonstrates that the cytoplasmic acidification is indeed causal for the tolerance. These new data can be found in Figs 5, S4K,L, S5A.

Since we made major changes covering large parts of the text, it would not result in a clear overview if we would fully cite all specific changes that we made for each of the comments listed below. Instead, we will indicate what we changed in response to each comment and refer to the line numbers of the revised manuscript without track changes where all the precise changes can be found when possible.

1. The first part of the manuscript, describing the evolution work and the resulting mutations, is difficult to follow. Clarification and simplification would help here, with more details to Materials and Methods.

As per suggestion of the reviewer, in addition to our answers to the specific comments below, we have adjusted the first part of the manuscript along with the Materials and Methods to clarify our description of the work.

Among the questions:

a. Why is UT189 a reference? What strain was actually used? Resource material says SX43, and references previous paper, but I could not find SX43 there. I gather this is a derivative of MG1655, since

that is probably the genome reference, but that is never clearly stated. Method as for 2016 with UT189? This was unclear. It was complicated because Fig. S1B is labeled NC_000093.3, which doesn't exist. Should this be NC_000913.3?

The UT189 is a widely used uropathogenic *E. coli* strain and was included to generalize our results obtained in the lab strain. This strain is now listed in Table S5 and we adjusted the methods section on lines 536-538 to make this clear.

The SX43 and SX2513 strains are the lab strains that we most frequently used in this study. They are close relatives to the BW25113 strain (ancestor of the Keio collection) containing a *venus* tag in the *lacZ* locus (published by Taniguchi et al, 2010 Science; we removed the KanR cassette in Van den Bergh et al, 2016 Nat Microbiol). These strains and all their descendants are now listed in the Table S5. We have now also added a short description to our methods to further clarify this (lines 536-538). The SX strains are further described in Van den Bergh et al, 2016 Nat Microbiol.

The evolution experiment was explained for the UT189 strain as the evolution trajectories for this strain have never been published. The experiment was carried out practically identically to the previously published evolution experiments of the wild-type strain. Therefore, we refer to that publication and explain the key points in this publication which allow replication of the experiment. We have added some extra information in the methods section, lines 600-6009 and 634-635.

We thank the reviewer for pointing towards the typo on the label of figure S1B. We have corrected that to NC_000913.3 as that was indeed the used reference.

b. Table S1 and elsewhere: Were mutations in noncoding regions not noted at all, or are there really none that reached the 5% level?

Only 3 out of a total of 128 mutations were found in non-coding regions above 5% in frequency. These 3 mutations can be found in Table S1. We have also added this information to the relevant methods section, lines 629-631.

c. Table S3 shows that there are more mutations, at lower frequencies, in the selected genes. I gather from the text that all 29 of the target genes listed in J, for instance, were examined for extra mutations. The legend for Table S3 should be very clear about what genes were examined and also whether there was any cutoff for frequency. It is not possible to tell from what was presented whether there are orders of magnitude more mutants everywhere, with some also in *nuo*.

We apologize for not having explained things properly. The reviewer is correct in that we only specifically looked at lower frequency mutations in the genes listed in Fig S1J. We actually expanded our analysis towards the entire operons of the identified target genes with mutations that reached the 5% level. For the analysis below 5% frequency, we did not apply any cut-off on frequency but instead, only kept very rudimentary quality filters, i.e., excluding repeat regions or mobile regions that are prone to sequencing/analysis errors. In regions beyond these target genes, we would also find low frequency mutations following this method but as such analysis is more error-prone, we restricted ourselves to analyse only the operons of the 29 of the target genes given our prior information. We now have added

more information to the legend of Table S3 and the methods on lines 624-628 to clarify these points.

d. Clarify what the carbon source was for growth in MHB media.

Mueller Hinton Broth is a complex, undefined medium similar to LB. As such, it contains only minor traces of sugars while it supplies peptides as carbon source (coming from casein hydrolysate and beef extract). This medium is widely used for antibiotic sensitivity testing in clinical and academic labs. We have added some more info on this medium to the Method section on line 548-549.

2. It would be very useful to understand to what extent the *nuo* alleles here are sufficient for the persistence phenotype (and other measured phenotypes). It should be possible, using the *yfbP::kan* linked marker (or another one), to transduce the mutant allele into the parental background. Does it should high persistence?

In almost all the clones that we sequenced, we only found 1-2 mutations (Table S2), which already is a strong indication for causality for those mutations. Additionally, we have cured mutants from their specific *nuo* mutation, using *yfbP::KmR* transduction from the wild type to the respective *nuo** mutant (the other way around as suggested by the reviewer). From the data now shown in Figs 2D and S2I, we conclude that the mutations in *nuoL*, *M* and *N* are sufficient for the persister phenotype, as these cured strains lose their antibiotic tolerance.

3. Lines 249-255, ATP:ADP ratios. This could be clearer. Maybe present the fact that ATP:ADP is down for both the points and the deletions, in both stationary (Fig. 4) and exponential (Fig. S4), and that this does not correlate with persistence. So low ATP:ADP is not sufficient to give high persistence; it doesn't really say that low ATP:ADP isn't necessary, does it? It would be useful to make that case clear, since it is, in my mind, qualitatively different from the data showing that antibiotic entry is unaffected. Here there is an effect, and it well may be important.

The reviewer is absolutely right, and we have corrected this section according to the suggestions. As we did not formally exclude the possibility that low energy contributes to the high persistence of the *nuo** mutants in this manuscript, we have adjusted the text to make sure that we do not make that claim. We now state that low energy cannot fully or single-handedly explain the high tolerance but can still contribute to persistence on lines 277-281.

4. In Figure 6, the authors use a number of different protocols to lead to persistence, and it is frequently difficult to follow to what extent these are likely to have the same pathways and components. Does a *nuo* mutant further increase persistence in the *hipA* allele? After switch from glucose to fumarate? Or are each of these working in the same pathway? Are the authors suggesting, from data like that in Fig. 5C, that in a shift from glucose to fumarate, the fraction of cells entering a persister state depends on them decreasing pH further/for a longer time, or is the lower pH over time the effect of entering a

persist state? What happens in an experiment like panel 5C, but with a *nuo* deletion that does not lead to high level persistence?

We agree with the reviewer that the *hipA7* data did not contribute a lot and actually could have caused some confusion. Since we wanted to keep focus on the *nuo*- and pH-dependent persistence mechanism and the microscopy data might have created some confusion, in an effort to further streamline the manuscript, we decided to exclude all the microscopy data. Furthermore, we decided to omit the mutants lacking *nuo* from the physiological measurements to tone down the complexity of the story. Furthermore, these strains show a very strong shifted metabolism making them inappropriate for comparison in these assays (see Fuhrer, 2017, MSB; Prüß, 1994, JBac). This became very obvious when we examined the global redox state in these strains, as seen in the figure below:

Here, resazurin reduction is followed in time (forming an increase in fluorescence). Mutants lacking *nuo* strongly differed in redox state from the wild type and *nuo** mutants that were highly similar, thus indicating a highly different metabolic state in mutants lacking *nuo* which would complicate using them as a comparison in physiological assays.

Considering the two setups used in our work, the entry in stationary phase in complex MHB medium and the switch from glucose to fumarate in exponential phase in minimal defined medium (done in two different labs): we would argue that, although it adds complexity to the story, it strengthens the value of the similar observations we make. Indeed, we suggest that the faster cells reach a lower pH_i during the switch, the more likely they are to become antibiotic tolerant. Our expanded dynamic pH_i measurements (Figs 5B-C and S5A) and tolerance assays of the wild type with acidified cytoplasm (Fig 5D) further strengthen our initial claims. We believe that this revised manuscript (with significant amount of new data added, and some less important data removed) is much more straightforward to follow. We feel that this does a great deal in addressing this reviewer's comment that things were difficult to follow.

5. If RpoS levels are increased in some other manner, is that sufficient to lead to persistence, as implied by some of the work here?

Indeed, it was previously shown by Radzikowski et al, 2016 MSB (Figure 7C) that overexpression of *rpoS* further increases persister levels upon a shift from glucose to fumarate, likely as more cells are being 'pulled' into persistence and less resources are available to restore metabolic homeostasis on fumarate. However, important to note, during such a shift and especially in mutants where pH_i homeostasis is further impaired (like the *nuo** mutants in this study), acidification still takes place and clearly affects the impact of RpoS. Our newly included results indicate that upon more severe acidification (such as in the *nuo** mutants), the RpoS dependency is largely bypassed and a full-blown stress response activated by RpoS - as is the case in the wild type (see Fig 6B and Radzikowski, 2016, MSB) - is not needed. The latter is concluded from the fact that there is still significant correlation between tolerance and pH_i in strains lacking *rpoS* and tolerance only decreases to a modest extent in *nuo** mutants lacking *rpoS*. This observation can be easily explained: For a RpoS stress response, protein expression is needed. If the cytoplasm acidifies more strongly (as in the *nuo** mutants) then the RpoS response cannot be "executed" as the protein expression came to a standstill. We have rephrased our results and discussion section to more specifically address the role of RpoS in the wild type and the *nuo** mutants (e.g. on lines 360-371).

6. Should Fig. S4J be labeled for internal pH?

We thank the reviewer and have added the label of internal pH as it must have been lost during the finalization of the figures.

7. The section of the paper on low pH and its relationship to persistence is not fully convincing. Among the issues:

a. In Fig. 6B, one cell that became a persister was followed, if I read this correctly, and a correlation of lower pH and increased RpoS was seen. This is interesting, but hardly evidence for the model suggested here. This is not in a *nuo* mutant; how clear is it that the same process goes on in the *hipA* allele? In any case, one cell obviously is not proof of much. No other cells express either lower pH or higher RpoS (are these always paired?).

The reviewer is right. Observing this cell was a "lucky shot" which we made by accident: Essentially, we observed a cell dynamically entering the persister phenotype: The cell stopped growing, its cytosolic pH dropped and it induced the *rpoS* promoter. Given the fact that persister cells are very rare, this was really a lucky observation. As these data are from a single cell, it is difficult to find many more of such cells and the microscopy data did not add much to our story (and perhaps caused some confusion), we opted to remove it from this manuscript. In addition, we have now also included additional evidence for a causal connection between persistence and pH_i (Figs. 5, S4 and S5).

b. The implication in the text is that the effect of *nuo* is via the lower pH and thus induction of RpoS. Have the authors ruled out changes in signaling via the ArcA/ArcB two component system, regulated by electron transport and known to affect RpoS levels? What is the evidence that it is lower pH that triggers the RpoS-dependent downstream effects?

We thank the reviewer for the thoughtful idea. In the light of our new experiments and analyses, we did not further investigate the role of ArcA/ArcB signalling on RpoS levels/activity for the following reasons. First, our new data confirms a significant correlation between tolerance and pH_i in strains lacking *rpoS* and only a slight decrease in tolerance in *nuo** mutants lacking *rpoS* (in Figs 6A-B, S4K-L and S5A). Therefore, the role of RpoS, especially in the context of the increased tolerance of the *nuo** mutants, seems to be modest. Secondly, we now also establish causality between internal acidification and an increased persister level (see Figs 5D and S5B), thereby strengthening our initial claims on the role of pH_i. Thirdly, our expanded dataset of proteome changes upon a glucose to fumarate shift indicates that the RpoS levels only increased at/right after the switch (where both wild type and mutants showed a similar acidification, see Fig 5B) and did not differ between the mutants and the wild type:

Likely, the further regulation of RpoS (activity) follows (only) complex posttranslational ways while the ArcA/B system is also reported to (post)transcriptionally affect RpoS levels (Battesti et al., 2011). Lastly, our supplemental Fig S6 already provides evidence on how a drop in pH_i as observed under our conditions, can affect RpoS levels, namely through the inhibition of SpoT-dependent and spontaneous ppGpp hydrolysis. While this provides a first specific mechanism to link pH_i directly with RpoS activity, it is mostly relevant for the wild type as in *nuo** mutants, ppGpp (and RpoS) play a more modest role (see above). We have adjusted our manuscript at various places to incorporate these new findings.

On the other hand and remarkably, if we look into ArcA expression in the proteomics data as suggested by the reviewer, it seems that the *nuo** mutants do not induce ArcA, whereas ArcA seems indeed induced in the wild type (see below). This matches both with the suggestion of the reviewer in that ArcA/B has a potential role for regulating RpoS activity in the wild type (in addition to SpoT/(p)ppGpp) and our observations that in *nuo** mutants, the RpoS dependency is largely bypassed by severe acidification and a full-blown stress response activated by RpoS (via ArcA/B) is not needed. Although these ArcA/B levels provide interesting indications for a potential role of ArcA/B in persistence, we believe a full and precise investigation of the role of ArcA/B to be out of scope for this already dense and extensive manuscript.

c. It seems important to examine the levels of RpoS, possibly with the RpoS::mcherry fusion, in the *nuo* cells.

We agree that RpoS levels could be of importance. To test the suggestion of the reviewer, we have determined the RpoS levels from our (expanded) proteomics data (see previous comment). Here we see that, besides for the delta RpoS strains where expression is at the detection limit, no major differences are found between strains. This is not entirely surprising given that a significant part of the regulation of RpoS has been described to occur post-translationally (Battesti et al., 2011) and given the modest importance of RpoS in the context of the *nuo** mutants (e.g. modest decrease in tolerance, still a negative correlation between tolerance and pH_i). We now more specifically describe the role of RpoS in in the revised version of the manuscript on lines 436-439 and 446-449.

d. In Fig. 6D, while it is clear that loss of *rpoS* has an effect in the *nuo* mutant on the proteome, what does the *rpoS* mutant alone do? It is a bit hard to evaluate the pattern here without that additional comparison. The Methods implies that after the fumarate switch, all cells are persisters. Is that the case? I could not find any of the actual proteins that are being monitored here. Is that all in previous publications? The primary data needs to be deposited somewhere.

In the revised manuscript we have greatly expanded the proteomics data to include all relevant strains and conditions. After the switch, most cells do switch to a persister state over time. In the wild-type strain, this happens over a much longer timer period (see also Radzikowski et al 2016 MSB) while the *nuo** mutants readily increase their persister fraction directly after the switch. Wild type lacking *rpoS* loses the ability to switch to the persister state (Fig. 6B), even after a longer time after the switch (see Radzikowski et al 2016 MSB), while *nuo** mutants lacking *rpoS* still switch significantly to the persister state (Fig. 6B). All these observations are in line with the differences in proteomic changes between these strains, as we now state in the revised manuscript. The proteome of *nuo** mutants with/without RpoS seems to “freeze” after the switch, whereas the wild type takes “the long road to persistence”

through the full activation of the RpoS stress regulon. A wild type without RpoS invests resources in restoring metabolic homeostasis, and not in massive activation of stress responses, which is also corroborated by a somewhat restored pH homeostasis (i.e., reversal of the initial acidification, Fig S5D) in the absence of RpoS in the wild type after a shift to fumarate, which explains its low persister level as this strain also has no protection (over time) from an activated RpoS general stress regulon.

The primary proteomics data that has been produced in this manuscript is made available on ProteomeXchange (ID: PXD029006) and will become publicly available upon publication, as was the case for datasets generated in our previous publications. The reviewer can access the data via the username: reviewer_pxd029006@ebi.ac.uk and password: g5Hk9OQa. The PCA spaces that were used here, were generated based on data of Radzikowski et al 2016 MSB, as we have now clarified further in the methods section on lines 848-853.

8. RelA and SpoT: It is somewhat surprising that deletion of *relA* and *spoT* doesn't affect persistence, given that persistence is RpoS dependent, and usually RpoS levels are significantly decreased in the absence of all ppGpp. That led me to look at the construction of these double mutants, which generally are also quite sick.

a. The Resource list says P1 was made on CF1693, but the listing on the *nuo relA* spot strains says they were transduced with P1 (JW5437-1), a strain that is not listed, and cured of kan. Which is it? How were the strains confirmed?

We are sorry for this mistake. We have clarified and corrected the resource list in Table S5 for the construction of these mutants as suggested. The *relA::KmR* cassette was obtained from strain **JW2755-1**, the Keio mutant, and the cassette was subsequently cured by FLP expression. The deletion of *spoT* was created using the CF1693 strain as the P1 donor. Here, the CmR cassette remained in place as it was not surrounded by FRT sites and therefore could not be readily removed. JW5437-1 was the donor strain to construct *rpoS* deletions.

All these strains were checked through PCR and size analyses using primers specified in Table S6. For the $\Delta relA \Delta spoT$ deletions, we also performed whole-genome re-sequencing (NCBI Bioproject: PRJNA768774). Here, we could confirm the intended deletions while no compensatory mutations could be discovered. Compensatory mutations are thus unlikely to underly the unchanged high antibiotic tolerance in the *nuo** mutants that lacked ppGpp. Furthermore, using the same genetics approach, we confirmed an important role of RelA (and SpoT) in the increased tolerance for other mutants selected by evolution (e.g., in *oppB*, *gadC*,...; unpublished confidential results). A loss of tolerance upon deleting *relA* and *spoT* in the *nuo** strains would indeed have been expected given the pH sensitivity of SpoT, but our results indicate that ppGpp has no role in the *nuo** strains. We have added the accession ID of the SRA files of the WGS on the *relA* and *spoT* deletion strains to the methods sections.

b. Were RpoS levels measured at all in these double mutants?

We did not do proteome analyses on the $\Delta relA \Delta spoT$ mutant or measured RpoS levels in another way as these deletions had little to no effect on the antibiotic tolerance.

c. If in fact SpoT and RelA are not necessary for the phenotype studied here, the section in the supplemental figures (Figure S5 and S6) on SpoT purification and effects of pH seems unnecessary. Why was SpoT included in Fig. 7, given the results here?

The reviewer is right. Given that for the *nuo**-dependent increased tolerance, RelA and SpoT seems to be irrelevant and the SpoT purification and effects of pH on ppGpp are indeed only a side-aspect of our story. However, given the logical connection of pH to RpoS via ppGpp and the remarkable pH sensitivity of SpoT, we do believe that this could be of importance in the wild-type strain or in conditions where acidification is less severe, RpoS levels are lower, ... Therefore, we prefer to include these remarkable results in a single supplemental figure. We have also added the rationale of a potential role of SpoT in the caption of figure 7. We would find it a pity if these interesting results would not be published.

9. The authors suggest, based on the experiments in Fig. 6D, E, and F, that a decrease in protein synthesis is an outcome of the *nuo*-based persistence. Is the decrease in leucine incorporation seen in 6E reversed in an *rpoS* mutant? Is it not seen in a *nuo* deletion?

We would like to refer to our previous comments on the modest role of RpoS and the fact that deletions of *nuo* are now omitted from the manuscript to tone down the complexity of the story. Furthermore, the leucine radioactive incorporation experiments are highly laborious and have become logistically difficult to repeat on additional strains due to changes in health, safety, and environment regulations. Nevertheless, using additional proteomics experiments on all these strains we now clearly confirm that changes in the proteome are absent in strains with strong acidification and high tolerance. These new data are now shown in Figure 6C.

10. Figure 7: In this figure, it would seem that the lower protein synthesis is independent of RpoS induction. That doesn't seem to be what the authors say in the text. Is the suggestion that some RpoS but not too much is necessary for persistence? What is the evidence for decreased translation inhibiting RpoS induction, or is that an assumption if all new translation is halted?

We agree with the reviewer and have further refined the description of our model in the revised manuscript, also based on the new additional data. Considering our new results, where $\Delta rpoS$ *nuo** strains show acidification and in fact only show modest decreases in tolerance, we have slightly adapted our model. The suggestion would indeed be the following: An RpoS response supports tolerance. However, an RpoS response, which requires synthesis of proteins, is only possible during a slow or/and limited acidification. If a fast and strong acidification occurs (as in the *nuo** mutants), then RpoS cannot exert its full effect. We explain this rationale in the new discussion, in the caption of Figure 7 and in the conclusion of our results on lines 436-439.

11. What is the evidence for halted protein degradation? If some degradation is going on, those amino acids might support synthesis of some proteins, not detected by the leucine labeling used here.

Next to leucine labelling experiments, we drew the conclusion of a halted protein synthesis/degradation also based on our proteome experiments. Here, we found that -remarkably- the change in the pattern of proteins comes to a complete stand still after 0.5-1 hours. We consider this – together with leucine experiment – very strong evidence that both proteins synthesis and degradation are downregulated. We make this rationale more clear in this revised version of the manuscript on lines 422-426.

12. Given the suggestion here that reduced translation is found in persistent cells, the authors may want to cite a recent paper finding that mutations in the translational apparatus give hyperpersistence (Khare and Tavazoie, 2020).

We appreciate bringing the paper to our attention. We have added this newly published results as part of our discussion on lines 518-519

Reviewer #4 (Remarks to the Author):

The paper by Heinemann and colleagues is aimed at providing mechanistic understanding of how perturbation of metabolic homeostasis leads to formation of a persister state. In my view there is no generic mechanism/trigger as many factors can trigger persistence.

We agree that persister formation likely is the result of a myriad of pathways or events, some acting together, some independent of each other and others contributing even under (slightly) different conditions. Nevertheless, our result on pH_i fits perfectly with this idea of many underlying specific pathways as pH_i can affect many (if not all) of the currently proposed factors that contribute to persistence.

The authors focus on the role of Complex I in triggering persistence in *Escherichia coli*. This is not a new finding and Roberto Kolter performed many studies in the early 1990s on GASP mutants (e.g. *JBact* 1993 175:5642-5647) and he uncovered complex I as being a pivotal player – not one of his papers is cited in this manuscript. I feel this is a major oversight by the authors and there needs to be some discussion about this.

We want to thank the reviewer for mentioning the interesting work of prof. Kolter. Indeed, he found that complex I is essential for certain GASP mutants to arise and take over the population in stationary phase, especially when GASP mutations are targeting *rpoS*, since, in these *rpoS* mutant backgrounds, *nuo* cannot be artificially knocked down without a loss of their GASP phenotype. Later, many other mutations in various pathways have been identified to be important for causing GASP (e.g. in *Irp*, *cpdA*, *rpoC*, ...) (e.g., Zinser, 2000, *JBac*; Chip, 2014, *mSphere*; Gonidakis, 2011, *Aging*) although the

contribution for complex I for these GASP mutants is less clear and no adaptive mutations in *nuo* were found to our knowledge.

However, nowhere in Roberto Kolter's work we could find anything that states the role of Complex I in triggering antibiotic persistence. We sincerely hope that we didn't overlook anything. The cited work or any of the works on GASPS mutants did not examine antibiotic resilience but measures fitness in stationary phase cultures that are free of antibiotics. Yet, to acknowledge the link with GASP, we now also included a short reference to the matter in the discussion section on lines 468-470.

The authors propose a model centred around cytoplasmic acidification through mutations in complex I (reduced proton pumping) inducing the persister state through the activation of the RpoS regulon and shutdown of protein synthesis. More experimental details are needed to fully explain this model.

For the revised version, based on the suggestions of all four reviewers, we have added much more experimental detail, included a significant number of additional experiments, analyses and textual changes that further underpin the main lines of our initially proposed model while specifying some of its details.

For instance, through re-measuring and expanding our (dynamic) measurements of pH_i and coupling these directly (in the same experiment) to the assessment of antibiotic-tolerant persister levels, we now find strong negative correlations between cytoplasmic pH (dynamics) and antibiotic tolerance levels (on all high-persistence *nuo** mutants with/without RpoS in two different setups across two different labs; see Figs 5C, S4K-L and S5A). Furthermore, through artificially modulating the intracellular pH, we now also establish causality between internal acidification and an increased persister level (see Figs 5D and S5B), thereby strengthening our initial claims. Lastly, we have expanded our experiments with dynamic pH_i measurements on all strains with/without *rpoS* while simultaneously measuring proteomes. These experiments confirmed that *nuo** mutants with strong acidification display a frozen proteome whereas modest acidification in the wild type triggers a full-blown stress response. Below, we explain in detail how we addressed the other comments of the reviewer.

Overall the paper is well written and presented and some interesting ideas are put forward by the authors. The experimental aspects of this manuscript are very impressive – a large number of high technical experiments have been performed. However, there are some important details missing that need clarification. I like the fact the authors are trying to come up with new mechanisms around persistence.

We would like to thank the reviewer for the appraisal of our work, for the effort that was put into reviewing our manuscript and for the constructive suggestions. Addressing the comments of the reviewer clearly improved the manuscript and made the story more streamlined and easier to understand.

Comments to address:

1. Lines 99-103: it would be good for the readers to understand why the antibiotics were chosen for this study. The focus was largely on inhibitors of protein synthesis – why?

The selection of protein inhibitors was a consequence of more practical preconditions: We were limited to bactericidal antibiotics as we wanted to study cell variants that can survive lethal doses of antibiotics. In addition, we work in stationary phase, which is a condition in which cells are notoriously more resilient towards lethal antibiotics. Lastly, we wanted to study persister cells. Therefore, obtaining a clear biphasic killing behaviour in the wild type at the start of the evolution experiments was imperative, as this would imply that only persisters are still viable and selected for the next round. Together, these factors limited our choices.

Furthermore, aminoglycosides are an important group of antibiotics used for treating severe bacterial infections and our initial trials of evolution under treatment with aminoglycosides showed clear results. In our work, we also included tolerance assays to another important class of antibiotics that is known to be bactericidal in stationary phase cultures, the fluoroquinolones (we used ofloxacin in Figs S2G-I, S4A and S4D). For this revised version, we now added the above arguments on lines 578-581.

2. The passaging protocol needs more details: For example, lines 551-555 – is the concentration of amikacin (400 ug/ml) sub-MIC, MIC or x-fold MIC? The same question applies to the other antimicrobials used in the study. What is the number of CFU/ml in the stationary phase culture for MHB? (Figure 1A)

The concentration is $\pm 100-200\times$ the MIC for amikacin. The MIC values of the strains and relevant antibiotics can now be found in Fig S2E. We added the missing concentrations of applied antibiotics and treatment times to the revised manuscript in both the captions of the relevant figures and the methods section.

CFU/ml in the stationary phase culture for MHB fluctuates between $1-5\times 10^9$ CFU/ml. This information is now added to the methods section and to the caption of Figure 1A.

3. Expression of the *nuo* operon. It has been reported that the *nuo* operon is regulated in response to various electron acceptors and oxygen availability. What is the pattern of *nuo* expression in the experimental setup used in this study to evolve mutants? For example, is the *nuo* operon expressed under these growth conditions (Fig. 1A)?

The reviewer is right that expression of *nuo* has been shown to be regulated by e.g., O_2 , electron acceptors and donors and acidity and is upregulated in stationary phase (e.g., Wackwitz, 1999, MGG; Salmon, 2003, Int. J. Biol. Chem.). Since we acquire a large number of point mutations in *nuo* and deletion of *nuo* abolishes the phenotype of high antibiotic tolerance, *nuo* needs to be expressed under the applied growth conditions as we would otherwise have no tolerance phenotype from these point mutations (which we proved to be causal).

A number of *nuo* mutants were characterised (lines 155-157) – do these *nuo* mutants show enhanced survival in the absence of antimicrobials under stationary phase conditions? Do these mutations occur naturally in the absence of antimicrobial stress – see work of Kolter above.

As per suggestion of the reviewer, we have included data on the long-term stationary phase survival of the *nuo** mutants below. Here, we measured the viable cell number over several days in stationary phase without any antibiotics in the rich complex MHB medium that was used during the evolution experiments. The survival of *nuoL*, *M* and *N* mutants and the wild type groups together and is lower than that of a *nuo* deletion strain. This test again shows the difference between a full *nuo* knockout that has no increased antibiotic tolerance and that we did not pick up as a result of our evolution experiments but was artificially created and used for comparison here and the subtle point mutations that we obtained. Furthermore, we have never observed the specific mutations that we study in the present manuscript, to be selected in absence of antibiotics. From this data, we conclude that long-term stationary phase survival in absence of antibiotic treatment (i.e. GASP phenotype) does not correlate with the high antibiotic tolerance that we study and, therefore, we decided to omit these data from the manuscript.

As explained earlier and to our best knowledge, Kolter and colleagues have never observed or selected any point mutations in *nuo*, let alone the precise type that we found (e.g., in the membrane spanning units, changing hydrophobic residues to hydrophilic ones). The only record of mutations in *nuo* in their work that we could find, are artificially created insertion (knockout) mutants that abolish the GASP phenotype of *rpoS* mutants. Therefore, the context and findings of the work of Kolter and colleagues is quite different from our current story. Yet, we now acknowledge the work of Kolter and colleagues on GASP and the role of *nuo* in *rpoS* GASP mutants on lines 468-471.

I was struggling to understand Fig. S1A - % survival is plotted in an unusual way in this Figure. Compare this to Figure 2D and 2E.

We have now made the % survival axes uniform throughout the manuscript using the short power of 10 notation to avoid the use of long decimal notations. Thanks for noting this point of inconsistency.

4. A key question in these *nuo* mutants is the expression of *ndh2* and cytochrome *bd* – do they switch to non-proton translocating complexes to maintain the pmf? Is there a change in the end product profile of the *nuo* mutants – can you detect fermentative end products?

In the newly added expanded proteomics dataset, we could detect expression of *ndh2* and *cytbd* (encoded by *cydAB*). As can be seen from the figure below, expression levels are very similar between any of the strains (except for a lower expression of *ndh* in strains lacking RpoS):

To check whether there is any change in the end product profile of the *nuo** mutants, we have obtained product profiles of the cultures growing on glucose before the shift to fumarate, analysed through HPLC and RI and DA detector signals (panel A and B in the figure below). As can be seen from the chromatographs, product profiles are highly similar between all strains and do not point towards an increased use of fermentative routes (e.g., no production of ethanol or pyruvate or fermentative products beside from the known overflow metabolism).

As there are no significant differences to report, we decided to omit these data from the revised manuscript.

5. Complex I (Figure 3). These experiments are very convincing but lack some important controls. I was unsure why the authors have not expressed the rates or NADH oxidation and oxygen consumption per

mg of protein? Are all the different complexes purified and reconstituted to the same level? Fig. 3A – what is the rate of NADH oxidation in the detergent micelle before reconstitution for the various complexes? Can these rates be calculated and reported.

We would like to thank the reviewer for pointing out the unclarity about the normalization and controls in this figure. As requested, In Fig 3A and 3B we have added insets with the rates of NADH oxidation and oxygen consumption per mg of protein, showing no significant changes between the different variant complexes. We have adjusted the text and caption of the figures and have also reported on the purification and reconstitution efficiencies and the control rates of NADH oxidation in the methods, e.g. on lines 756-760.

6. Figure 3B: I am assuming this is inside-out vesicles (IMV) and therefore the rate should be reported per mg protein. If it is proteoliposomes, I would like to see the IMV data.

This experiment was performed on membranes for which protein concentration was determined by the biuret method. We have now added this information in the caption and added the rates per mg protein in an inset. Concentrations were similar between the variants and therefore rates were highly comparable and amounted to an NADH/oxidase activity of 0.16 U/mg, in the range of values found in literature (added to the caption of Fig. 3).

Monitoring O₂ reduction by complex I variants in proteoliposomes is impossible as terminal oxidases are lacking in such a setup. In this work, proteoliposomes and inside-out vesicles were specifically used to probe H⁺ translocation (in Figs 3C and D).

7. Figure 3C: shows that the proteoliposomes are leaky as the quench is reversed with time. This is not observed in the IMVs (Fig. 3D) – a control needs to be included to dissipate the delta pH to prove indeed it has been generated – could add CCCP. The data does indeed show impaired proton pumping but how does that compare with rates of NADH oxidation – are these mutants simply uncoupled i.e. some proton leak in the complex?

The reviewer is absolutely correct, proteoliposomes with reconstituted complex I are not as tight as the IMVs. Nevertheless, the traces shown in Fig. 3C and 3D do rightfully reflect the generation of a proton gradient by complex I because additional control experiments with CCCP and a complex I inhibitor, as requested, prove that ΔpH is generated, specifically by complex I (included as Figs. S3G-H.)

We now also mention in the caption of Fig 3 that the proteoliposomes are leaky and that the IMVs do not leak as these membrane vesicles are much tighter than the artificially made proteoliposomes. Furthermore, the IMV contain the terminal quinol oxidases that are also translocating protons. We agree with the statement of the reviewer that the mutations in complex I lead to a partial uncoupling of proton translocation and electron transfer in complex I and specifically mentioned the statement of the reviewer in the text on lines 219-220.

8. Line 239/Figure 4: it would be good to have some quantitative data on the membrane potential measurements. Same comment applies to delta pH measurements. The authors use CCCP as a control in Figure 4 – what happens when nigericin is used i.e. can you convert any delta pH into the membrane potential? How do the authors explain the increased uptake of amikacin in the 294D mutant (Fig 4B) – is the membrane hyperpolarised in this mutant? What is the total pmf in this mutant?

We agree with the reviewer that a full detailed and quantitative image of membrane potential would be interesting. However, given the large body of new results and the already complex story as it is, we believe that a quantitative measurement of membrane potential is beyond the scope of the current article. Our contacts with an independent expert in the field (prof. Bert Poolman of Groningen University) confirmed that precise quantitation of $\Delta\Psi$ are highly complex, especially in living bacteria, and technically highly demanding (e.g. requiring radioactive TPP⁺ ions, various controls with compounds like nigericin that not easily pass the outer membrane of Gram negative bacteria) and we believe that our more qualitative measurements of potential with the fluorescence of DiBAC₄(3) sufficiently substantiates our statements. Instead, we opted to change Fig 4A, by displaying cytometer data as 1-cumulative distributions as per suggestion of another reviewer to improve comparison, thereby revealing a minor depolarization in the *nuo** mutants. We have added a quantitative, biological context to these small differences in the text (lines 252-261) and caption of Fig 4 to explain why such small differences (that furthermore do not lead to decreased uptake of antibiotic), do not explain the large differences in persistence levels observed in these strains. Certainly, no hyperpolarization is observed that could explain the higher uptake of the L variants. We currently cannot explain this increased uptake of the L variant but the goal of the assays in Fig 4A-B was to check for a lower electrical potential and antibiotic uptake, which we can exclude on these data as a mechanism for tolerance.

For quantitative Δ pH measurements, we would like to refer the reviewer to our substantially expanded (dynamic) measurements of cytoplasmic pH in the revised manuscript, which we furthermore directly correlated (in the same experiment) to antibiotic tolerance (in two different setups across two different labs; see Figs 5C, S4K-L and S5A). As such, we did not only find strong negative correlations between cytoplasmic pH and antibiotic tolerance levels but also identified a causal link between internal acidification and an increased persister level, thereby strengthening our initial claims.

9. pH homeostasis Figure 5: a key piece of information missing in Figure 5 is that of the external pH values under the different conditions. This is crucial information, especially in the cells challenged with fumarate.

For the experiments in minimal medium (either glucose or fumarate), medium was set at pH 7 at the start, and we found the pH to stay stable throughout the experiment. For the experiments in complex, MHB medium the pH was initially 7.12 but alkalized for all strains upon entry in stationary phase to a value 8.1 ± 0.06 . These values are now specified in figure captions or the methods section.

E. coli is able to tolerate huge changes in intracellular pH with no effect on cell physiology. It has a battery of mechanisms to combat extreme intracellular acidity. The extremely modest changes in intracellular pH observed here are very difficult to reconcile when any profound effects on cell

physiology. For example the “acidic” pH values are still around intracellular pH 7.0. This internal pH would have little effect on protein synthesis.

While we appreciate the reviewer’s critical mind, we strongly disagree with this statement. To our knowledge, *E. coli* is known to tolerate huge changes in extracellular pH and this, depending on the extremity of the extracellular acidity/alkalinity, with sometimes quite extensive effects on physiology as they stop growth and have to reroute metabolism to counter intracellular acidification/alkalination (indeed pH_i homeostasis is an extremely tightly controlled process, e.g., see Krulwich, 2011, NRM; Kanjee, 2013 Ann. Rev. Microbiol). We have found no reports where *E. coli* displays huge changes in intracellular pH, let alone where this is the case and physiology is not affected and strains remained antibiotic sensitive.

As can be seen in the revised manuscript and in our replies to previous comments, we have now substantially expanded our measurements of cytoplasmic pH with relation to antibiotic tolerance. As such, we did not only find strong negative correlations between cytoplasmic pH and antibiotic tolerance levels but also identified a causal link between internal acidification and an increased persister level, thereby strengthening our initial claims. While perhaps perceived as modest changes, pH is expressed on a log scale, so a difference of “only” 0.5 pH units already represents a 3.2-fold change in H⁺ activity. Furthermore, it is important to realize that internal pH (dynamics) are strictly regulated in *E. coli* (and other organisms) and are tailored to (changes in) growth conditions. Modest deviations from such strict regulatory process with furthermore altered dynamics happening in conditions without any external pH stress (as we observe in the high-persistence conferring *nuo* mutants), can therefore nevertheless have pervasive effects. Our recorded values are not acidic as such, but rather acidified compared to the norm (and the external pH; we now made sure that we do not describe the cytoplasm as acidic throughout the text), which for example already strongly inhibits the ppGpp hydrolysis rate of SpoT (see Fig S6) and can affect stability, charge, folding state, mobility, redox equilibria, activity, interaction, ..., more broadly of many proteins and metabolites at the same time (e.g.: Orij, 2011, BBA; Kozłowski, NAR, 2017).

Throughout various systems in life, modest changes in pH_i (±0.5 or lower) were also shown to have significant effects, e.g.:

- strongly affect the O₂ saturation of hemoglobine in our blood (the Bohr/Haldane effect (Bohr, 1904, Skand. Arch. Physiol.; Christiansen, 1914, J. Physiol.),
- disrupt hamster embryo development (Squirrel, 2001, Biol. Reprod.), impede starfish oocytes meiosis (Hosada, 2019, JCB), controls WNT signaling in amniote embryos (Oginuma, 2020, Nature)
- prevent excitotoxic and ischemic neuronal death in the context of stroke, brain trauma, and neurodegenerative disorders (Lam, 2013, PNAS),
- regulate cancer and stem cell behavior (e.g., Persi, 2018, Nat Comm.; for a review article: Liu, 2020, Front Oncol),
- have marked effects on neuronal Ca²⁺ signaling (e.g., Willoughby, 2001, J. Physiol.; Križaj, 2011, Am J Physiol Cell Physiol; Gavriliouk, 2017, Sci Rep)
- be crucial in the functioning of the secretory pathway in plants and animals (e.g., Paroutis, 2004, Physiology; Linders, 2021, biorxiv; Martinière, 2013, The Plant Cell)
- modulate autophagy and mitophagy (Berezhnov, 2016, JBC) which underlies neurodegenerative diseases, most notably Parkinson disease.

- explain chemoresponse to NH_4^+ in *Paramecium* (Davis, 1998, Cytoskeleton) and influence bacterial thermotaxis (Demir, 2012, Biophys. J.),
- act as a secondary messenger involved in glucose-sensing in yeast (Isom, 2018, JBC) where it links membrane biogenesis to metabolism (Young, 2010, Science) and controls the growth rate (Orij, 2012, Genome Biol.)
- ...

In this context and given our further expanded dataset, we believe our claims that pH_i can contribute to antibiotic tolerance to be justified.

If these mutants were truly defective in pH homeostasis an experiment should be performed in which the mutants are challenged with acidic pH in the range pH 5-7 and the internal pH measured. This would reveal a true defect in pH homeostasis – see PMID:11283297. One would argue that exposing wild-type cells to acidic pH to lower internal pH should also trigger persistence. Of course this would also reveal hyperpolarization of the membrane potential as the cells would lack the ability to interconvert the membrane potential into a delta pH (compromised proton pumping).

We thank the reviewer for the thoughtful suggestion. According to this request, we have now included in our revised manuscript experiments where we resuspend wild-type cells in spent medium that was adjusted at various pH values. Here we found, fully consistent with our claim, that when we acidify the cytoplasm using a weak acid/base couple, we induce antibiotic tolerance (Fig. 5D), thereby providing strong indications for a causal relationship. The *nuo** mutants decrease their antibiotic tolerance somewhat when resuspended in acidified spent medium (we assume, as the reviewer suggested, because of pleiotropic costs due to defective pH homeostasis) but they also restore their tolerance as soon as the weak acid/base couple is included (Fig. S5B).

10. Figure 5: I need a clearer justification of the rationale for including fumarate in these experiments – is it being used as a carbon source or electron acceptor, which would imply hypoxia. What is the differences in extracellular acidification between glucose and fumarate? Have the authors tried any experiments under hypoxia – a stationary phase cell would be fairly hypoxic in this experimental setup. The authors might want to consider the review of Voskuil in their discussion (PMID: 30262111). Perhaps the alkaline pH in the wild-type explains the lack of tolerance?

In the experiments involving a glucose-fumarate switch, fumarate is used as carbon source and oxygen is the electron acceptor. This can be seen from our earlier report, where we found that the persister cells on fumarate take up oxygen (Fig. 2D in Radzikowski et al, 2016, Molecular Systems Biology). As mentioned in a previous comment, the pH of the minimal medium used for the glucose-fumarate shift, was set at 7 at the start, and stayed at 7 for the duration of the shift on which we report in our manuscript. In an exponentially growing culture on glucose, the pH typically drops due to acetate production.

As for the statement on “Perhaps the alkaline pH in the wild-type explains the lack of tolerance”: Yes, this is highly similar to the point that we want to make in our manuscript. The *nuo** mutants are protected from killing as the intracellular pH drops, which does not happen in the wild type leading to its killing. All the newly added data further strengthen this causal connection. Of note, in contrast to the group of Voskuil and their work on *Mycobacterium*, we do not see alkalinisation of the wild type during treatment, pointing towards a subtle difference between the data of the Voskuil group and our present

data. To acknowledge the interesting work of Voskuil and colleagues, we have now added a short section to the discussion on lines 452-455 and 495-496.

REVIEWER COMMENTS

Reviewer #1 (Remarks to the Author):

The authors have done a good job in addressing the comments raised in our original review. We recommend the revised paper for publication in Nature Communications.

Reviewer #2 (Remarks to the Author):

The authors have done a major revision of their previous manuscript, repeating experiments and leaving out the weaker data sets. The work is very interesting and a valuable addition to the ongoing search for persister cells mechanisms.

Reviewer #3 (Remarks to the Author):

This revised manuscript presents a quite thorough analysis of the ways in which point mutations in the *nuo* complex lead to high levels of persistence. The primary finding is that a change in intracellular pH in stationary phase is necessary for the persistence and that this leads to a shut-down in translation (by mechanisms still to be determined). While RpoS contributes to the phenotype, it is not a major component. The revisions make this complicated story easier to follow, and address most of the issues raised by reviewers. Given the continuing interest in bacterial persistence, this work provides important insights into this process.

Some minor points needing to be fixed:

Line 414: No red square seen in Fig. 6D.

Line 428: Fig. 6D referenced here, but doesn't seem to be *rpoS*- strains (or doesn't say so). Is there some other figure that this is supposed to reference?

Reviewer #4 (Remarks to the Author):

I am pleased to see the authors have made a thorough effort in revising this manuscript. I am satisfied with the rebuttals I have read through. I have no further comments.

REVIEWERS' COMMENTS

Reviewer #1 (Remarks to the Author):

The authors have done a good job in addressing the comments raised in our original review. We recommend the revised paper for publication in Nature Communications.

We are happy to see that our revisions are highly appreciated by the reviewer.

Reviewer #2 (Remarks to the Author):

The authors have done a major revision of their previous manuscript, repeating experiments and leaving out the weaker data sets. The work is very interesting and a valuable addition to the ongoing search for persister cells mechanisms.

We are happy to see that our revisions are highly appreciated by the reviewer.

Reviewer #3 (Remarks to the Author):

This revised manuscript presents a quite thorough analysis of the ways in which point mutations in the nuo complex lead to high levels of persistence. The primary finding is that a change in intracellular pH in stationary phase is necessary for the persistence and that this leads to a shut-down in translation (by mechanisms still to be determined). While RpoS contributes to the phenotype, it is not a major component. The revisions make this complicated story easier to follow, and address most of the issues raised by reviewers. Given the continuing interest in bacterial persistence, this work provides important insights into this process.

We are happy to see that our revisions are highly appreciated by the reviewer.

Some minor points needing to be fixed:

Line 414: No red square seen in Fig. 6D.

Thank you for pointing out this mistake that arose during the revisions. We now corrected the reference to the red dot in Fig. 6c

Line 428: Fig. 6D referenced here, but doesn't seem to be rpoS- strains (or doesn't say so). Is there some other figure that this is supposed to reference?

Thanks for detecting this mistake, we wanted to refer to Fig. 6c and have now corrected this

Reviewer #4 (Remarks to the Author):

I am pleased to see the authors have made a thorough effort in revising this manuscript. I am satisfied with the rebuttals I have read through. I have no further comments.

We are happy to see that our revisions are highly appreciated by the reviewer.